# 53BP1 and BRCA1 control pathway choice for stalled replication restart

**Yixi Xu[1†], Shaokai Ning[1†], Zheng Wei[1], Ran Xu[1], Xinlin Xu[1], Mengtan Xing[1,2], Rong Guo[1]\*, Dongyi Xu[1]\***

[1]State Key Laboratory of Protein and Plant Gene Research, School of Life Sciences, Peking University, Beijing, China; [2]Department of Cancer Research and Molecular Medicine, Norwegian University of Science and Technology, Trondheim, Norway

**Abstract** The cellular pathways that restart stalled replication forks are essential for genome stability and tumor prevention. However, how many of these pathways exist in cells and how these pathways are selectively activated remain unclear. Here, we describe two major fork restart pathways, and demonstrate that their selection is governed by 53BP1 and BRCA1, which are known to control the pathway choice to repair double-strand DNA breaks (DSBs). Specifically, 53BP1 promotes a fork cleavage-free pathway, whereas BRCA1 facilitates a break-induced replication (BIR) pathway coupled with SLX-MUS complex-mediated fork cleavage. The defect in the first pathway, but not DSB repair, in a 53BP1 mutant is largely corrected by disrupting BRCA1, and vice versa. Moreover, PLK1 temporally regulates the switch of these two pathways through enhancing the assembly of the SLX-MUS complex. Our results reveal two distinct fork restart pathways, which are antagonistically controlled by 53BP1 and BRCA1 in a DSB repair-independent manner.
DOI: https://doi.org/10.7554/eLife.30523.001

**\*For correspondence:**
guorong@pku.edu.cn (RG);
xudongyi@pku.edu.cn (DX)

[†]These authors contributed equally to this work

**Competing interests:** The authors declare that no competing interests exist.

## Introduction

Damaged (stalled or collapsed) forks are a major cause of genome instability in tumorigenesis (*Burrow et al., 2009*). Cells possess robust pathways to restart damaged forks (*Franchitto et al., 2008*; *Neelsen and Lopes, 2015*; *Petermann and Helleday, 2010*). Broadly, these pathways can be divided into two main pathways based on their distinct mechanisms. In the first pathway, the stalled forks are stable, and no breakage is generated during the restart process. The structures of these forks might undergo remodeling, which may include re-annealing of excess single-strand DNA (ssDNA) and/or regression to form a Holliday junction-like intermediate (*Neelsen and Lopes, 2015*). In contrast, the second pathway is fork cleavage-coupled BIR. BIR has been studied extensively in budding yeast as a homologous recombination (HR)-mediated repair pathway for broken forks after collapse (*Anand et al., 2013*; *Malkova and Ira, 2013*; *Mayle et al., 2015*). However, several lines of evidences have shown that the breakage of stalled forks is not the consequence of unprogrammed collapse but an active process mediated by the MUS81 endonuclease, which promotes fork restart, particularly after prolonged replication stress (*Hanada et al., 2007*; *Pepe and West, 2014*; *Shimura et al., 2008*). These findings imply that BIR may not only passively repair broken forks, but also actively restart stalled forks by coupling with MUS81-mediated cleavage. Recently, the scope of MUS81-coupled BIR was expanded to DNA replication repair in mitosis (*Bhowmick et al., 2016*; *Minocherhomji et al., 2015*; *Sotiriou et al., 2016*). However, the importance of the MUS81-mediated pathway has been questioned by studies using human cancer cell lines, as several independent studies have shown that DSBs formed in response to replication stress play no roles in fork restart in these cells (*Franchitto et al., 2008*; *Petermann and Helleday, 2010*; *Petermann et al., 2010*). It remains unclear how these mutually exclusive pathways are regulated in cells and how a stalled fork makes its choice of which pathway to activate.

53BP1-RIF1 and BRCA1-CtIP antagonistically control the pathway choice between non-homologous end joining (NHEJ) and HR for DSB repair by determining resection of broken ends (*Bunting et al., 2010*; *Escribano-Díaz et al., 2013*). Here, we find that 53BP1 and BRCA1 have similar antagonistic interactions in governing the fork restart pathways. In this case, the choice of which fork restart pathway to activate is determined by the cleavage of the stalled forks. BRCA1 promotes programmed cleavage and thus supports the cleavage-coupled BIR pathway, whereas 53BP1 antagonizes BRCA1-dependent cleavage and thus supports the cleavage-free pathway. Our data indicate that 53BP1 and BRCA1 antagonize each other not only to control DSB repair pathways, but also to restart stalled forks using a novel mechanism.

## Results

### 53BP1 and RIF1 have a NHEJ-independent function in response to replication stress

Our previous study showed that RIF1 promotes stalled replication restart and *RIF1*-deficient DT40 cells are hypersensitive to the replication inhibitors hydroxyurea (HU) and aphidicolin (APH) (*Xu et al., 2010*). We found that both *53BP1*$^{-/-}$ and *RIF1*$^{-/-}$ DT40 cells are hypersensitive to HU and APH (*Figure 1A*). Importantly, *53BP1*$^{-/-}$/*RIF1*$^{-/-}$ double knockout cells showed similar sensitivity as the single knockout cells (*Figure 1A*), demonstrating that 53BP1 and RIF1 are in the same pathway for cellular resistance to replication stress.

To examine whether the functions of 53BP1 and RIF1 in the cell response to replication stress are due to their role in NHEJ repair, we examined the function of Ku70. In the NHEJ pathway, Ku70 is a core factor and is more essential than 53BP1 and RIF1, which are regulators (*Escribano-Díaz et al., 2013*). Consistently, compared with *53BP1*$^{-/-}$ or *RIF1*$^{-/-}$ cells, *Ku70*$^{-/-}$ cells were much more sensitive to ICRF193 (*Figure 1B*), which is a Topo2 inhibitor and whose sensitivity is a widely used readout of NHEJ-deficiency (*Adachi et al., 2003*; *Wang et al., 2010*; *Xing et al., 2015*). Moreover, we examined random integration efficiency, which mainly depends on the NHEJ pathway in DT40 cells (*Escribano-Díaz et al., 2013*). *Ku70*$^{-/-}$ cells showed a much lower integration efficiency (approximately 300-fold less than the wild-type cells) than did the *53BP1*$^{-/-}$ and *RIF1*$^{-/-}$ cells (approximately 5-fold reduction; *Figure 1C*). Conversely, the *Ku70*$^{-/-}$ cells showed very weak or no sensitivity to HU and APH (*Figure 1B*), suggesting that defect in the NHEJ pathway is unlikely to account for the cellular sensitivity to replication stress. Thus, the functions of 53BP1 and RIF1 in response to replication stress are independent of their roles in NHEJ.

### The absence of BRCA1 suppresses the hypersensitivity of 53BP1-deficient cells to replication stress

*BRCA1*$^{-/-}$ DT40 cells were not only sensitive to PARP inhibitor (Olaparib) and topoisomerase I inhibitor camptothecin (CPT), but also sensitive to HU (*Figure 2—figure supplement 1A*). These sensitivities were rescued by re-introducing wild-type human BRCA1 or I26A mutant, which loses ubiquitin ligase activity, but not C61G mutant, which loses both ubiquitin ligase activity and its interaction with BARD1(*Ruffner et al., 2001*) (*Figure 2—figure supplement 1A and B*). Olaparib- and CPT-induced DNA damages require BRCA1-dependent HR for repair (*Bunting et al., 2010*). These results suggest that its interaction with BARD1 but not ubiquitin ligase activity is important for functions of BRCA1 in response to replication stress and in HR. Interestingly, BRCA1 mutant M1775R, which localizes in its BRCT domain and disrupts its interaction with CtIP, FANCJ and RAP80 (*Huen et al., 2010*), rescued the HU- but not Olaparib- or CPT-sensitivity of the *BRCA1*$^{-/-}$ cells (*Figure 2A* and *Figure 2—figure supplement 1C*), suggesting that the function of BRCA1 in response to replication stress is distinct from its role in HR.

Surprisingly, the sensitivity of *53BP1*$^{-/-}$ cells to replication stress was strongly suppressed by the disruption of the *BRCA1* gene (*Figure 2B* and *Figure 2—figure supplement 1D*). In fact, *53BP1*$^{-/-}$/*BRCA1*$^{-/-}$ cells were even more resistant to HU than were *BRCA1*$^{-/-}$ cells. This genetic interaction suggests that 53BP1 and BRCA1 are in the same pathway and counteract each other in response to replication stress. Then we tested whether this antagonistic function is due to their counteracting function in DSB repair. We found that the NHEJ-defect phenotypes, ICRF193-sensitivity and decreased random integration efficiency, of *53BP1*$^{-/-}$ cells were not recovered when *BRCA1* was

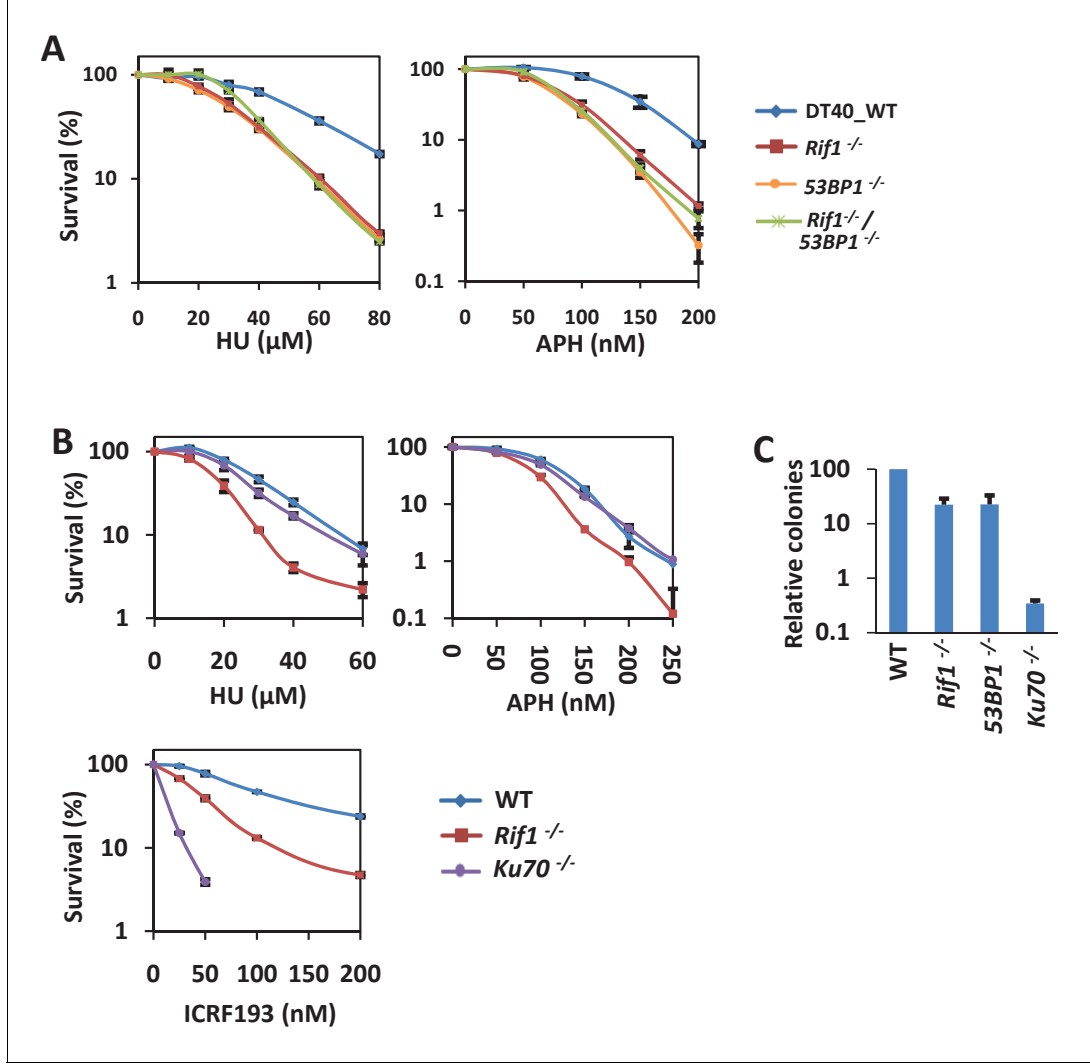

**Figure 1.** 53BP1 and RIF1 play a role in resisting replication stress in a DSB repair-independent manner. (A) Genetic interaction analysis between RIF1 and 53BP1 by sensitivity assay using MTT staining in DT40 cells. The mean and s.d. from three independent experiments are shown. (B) Sensitivity assay of variant DT40 cells. The mean and s.d. from three independent experiments are shown. (C) Random integration assay of variant DT40 cells. The mean and s.d. from three independent experiments are shown.

DOI: https://doi.org/10.7554/eLife.30523.002

disrupted (*Figure 2—figure supplement 1D and E*). Thus, we concluded that 53BP1 and BRCA1 have novel functions in response to replication stress that are independent of their function in DSB repair.

Additionally, $RIF1^{-/-}/BRCA1^{-/-}$ cells were not more sensitive to HU and APH than either of the single knockout cells (*Figure 2—figure supplement 1F*), suggesting that RIF1 and BRCA1 are also in the same pathway for cellular resistance to replication stress. However, the replication stress-sensitivity of $RIF1^{-/-}$ cells was not as well rescued as that of $53BP1^{-/-}$ cells when BRCA1 was disrupted. Similar phenomenon has been observed for repair of DSBs, in which the absence of 53BP1 rescued HR in only BRCA1-deficient cells, but not CtIP or XRCC2–mutant cells (*Bunting et al., 2010*; *Escribano-Díaz et al., 2013*). Therefore, we hypothesized that 53BP1 is an upstream regulator; while RIF1 is a downstream factor which may play an essential role in the pathway, such as recruiting BLM to stalled forks (*Xu et al., 2010*).

To assess whether this phenomenon also exists in mammalian cells, we generated $53BP1^{-/-}/BRCA1^{-/-}$ double knockout HCT116 cells using CRISPR (*Figure 2—figure supplement 2A–F*). Similar

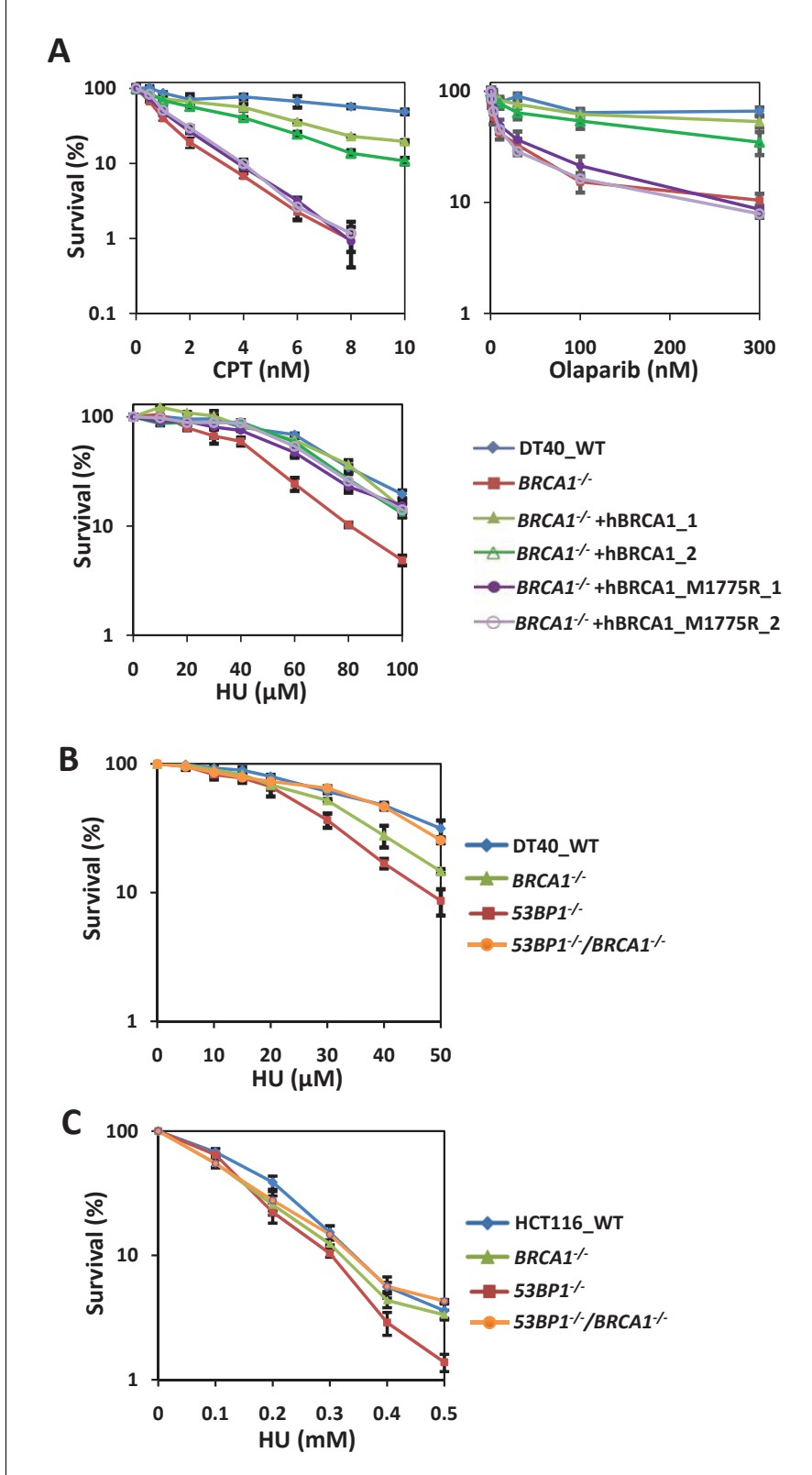

**Figure 2.** BRCA1 and 53BP1 interact antagonistically to resist replication stress in a DSB repair-independent manner. (**A**) Sensitivity assay of *BRCA1⁻/⁻* DT40 cells complemented with wild type or M1775R human BRCA1. The mean and s.d. from three independent experiments are shown. (**B**) Genetic interaction analysis between BRCA1 and 53BP1 by sensitivity assay using MTT staining in DT40 cells. The mean and s.d. from three independent

*Figure 2 continued on next page*

*Figure 2 continued*

experiments are shown. (C) Sensitivity assay of wild-type, *53BP1^-/-^*, *BRCA1^-/-^* and *53BP1^-/-^ BRCA1^-/-^* HCT116 cells. The mean and s.d. from three independent experiments are shown. Please refer to *Figure 2—figure supplement 1* and *Figure 2—figure supplement 2* for additional information in support of *Figure 2*.

DOI: https://doi.org/10.7554/eLife.30523.003

The following figure supplements are available for figure 2:

**Figure supplement 1.** BRCA1 and 53BP1 interact antagonistically to resist replication stress in a DSB repair-independent manner.

DOI: https://doi.org/10.7554/eLife.30523.004

**Figure supplement 2.** Generation of BRCA1 and 53BP1 knockout HCT116 cells.

DOI: https://doi.org/10.7554/eLife.30523.005

to DT40 cells, *53BP1^-/-^* HCT116 cells were sensitive to replication inhibitor, and this phenotype was rescued by the disruption of the *BRCA1* gene (*Figure 2C*), suggesting that the antagonistic functions of 53BP1 and BRCA1 in response to replication stress are conserved in vertebrates.

## 53BP1 and BRCA1 promote the fast and slow kinetics restart pathways, respectively

Replication stress sensitivity usually results from defects in the restart of stalled forks. Therefore, we examined the replication progression rate under conditions of low replication stress (0.1 mM, 0.2 mM or 0.5 mM HU). All these cells showed a significant decrease in IdU tract lengths during HU exposure (*Figure 3A*). However, *53BP1^-/-^* cells exhibited a larger decrease in the progression rate upon HU treatment than wild-type cells (*Figure 3A*). Therefore, these results suggest that 53BP1 is required for replication progression under stress. In comparison, *BRCA1^-/-^* and *BRCA1^-/-^/53BP1^-/-^* cells displayed similar replication tracts as wild-type cells (*Figure 3A*). Thus, the loss of BRCA1 promotes replication progression in 53BP1-deficient cells, which correlates with HU sensitivity.

We then directly examined the restart efficiency after replication stalling with high concentrations of replication inhibitors. We first tested the fast restart ability of wild-type and mutant cells after short-term (20 min) recovery following different times of replication inhibition (*Figure 3B*). The fast restart ability was reduced across all genotypes when the exposure time to replication stress was prolonged. However, compared with wild-type cells, 53BP1-deficient cells had significantly decreased restart efficiencies at almost all treatment times, suggesting that 53BP1 is required for a fast-kinetics pathway of restart. In comparison, *BRCA1^-/-^* cells showed comparable fork recovery abilities to wild-type cells, suggesting that BRCA1 is not required for this pathway. Moreover, *BRCA1^-/-^/53BP1^-/-^* cells displayed a significantly higher restart efficiency than *53BP1^-/-^* cells, suggesting that BRCA1 suppresses the fast restart pathway in 53BP1-deficient cells, consistent with sensitivity assay.

We also tested the restart efficiency after prolonged recovery times (20 min, 40 min and 60 min) following a medium-length period (12 hr) of replication inhibition (*Figure 3C*). The restart efficiencies in all cell lines, except *BRCA1^-/-^* cells, were increased when the recovery time was prolonged (from 20 min to 40 min and 60 min), suggesting that BRCA1 is required for a slow-kinetics pathway. Both wild-type and M1775R BRCA1 rescued the defect of the slow-kinetics pathway in the *BRCA1^-/-^* cells (*Figure 3—figure supplement 1A,B*), indicating that the function of BRCA1 in fork restart is independent on HR. *53BP1^-/-^* cells displayed a comparable restart rate to wild-type cells after prolonged recovery times (40 min and 60 min), suggesting that the slow-kinetics fork restart pathway is not impaired in 53BP1-deficient cells. Although the percentage of restarting forks of the *53BP1^-/-^* cells is recovered to a level similar as that of wild-type cells when recover time is prolonged to 40 min, the length of the restarted tracks is significantly shorter (*Figure 3D*), suggesting that the activated BRCA1-dependent pathway in the *53BP1^-/-^* cells is a delayed mechanism of fork restart. Moreover, the absence of 53BP1 promotes the slow-kinetics fork restart pathway in *BRCA1^-/-^* cells (*Figure 3C and D*), consistent with the replication stress sensitivity assay.

## The 53BP1- and BRCA1-dependent pathways mainly works in early and late stages of replication inhibition, respectively

The fast-kinetics fork restart pathway (recovered in 20 min) of wild-type HCT116 cells was very efficient after a short time of replication inhibition (57% at 2 hr), but it progressively decreased over

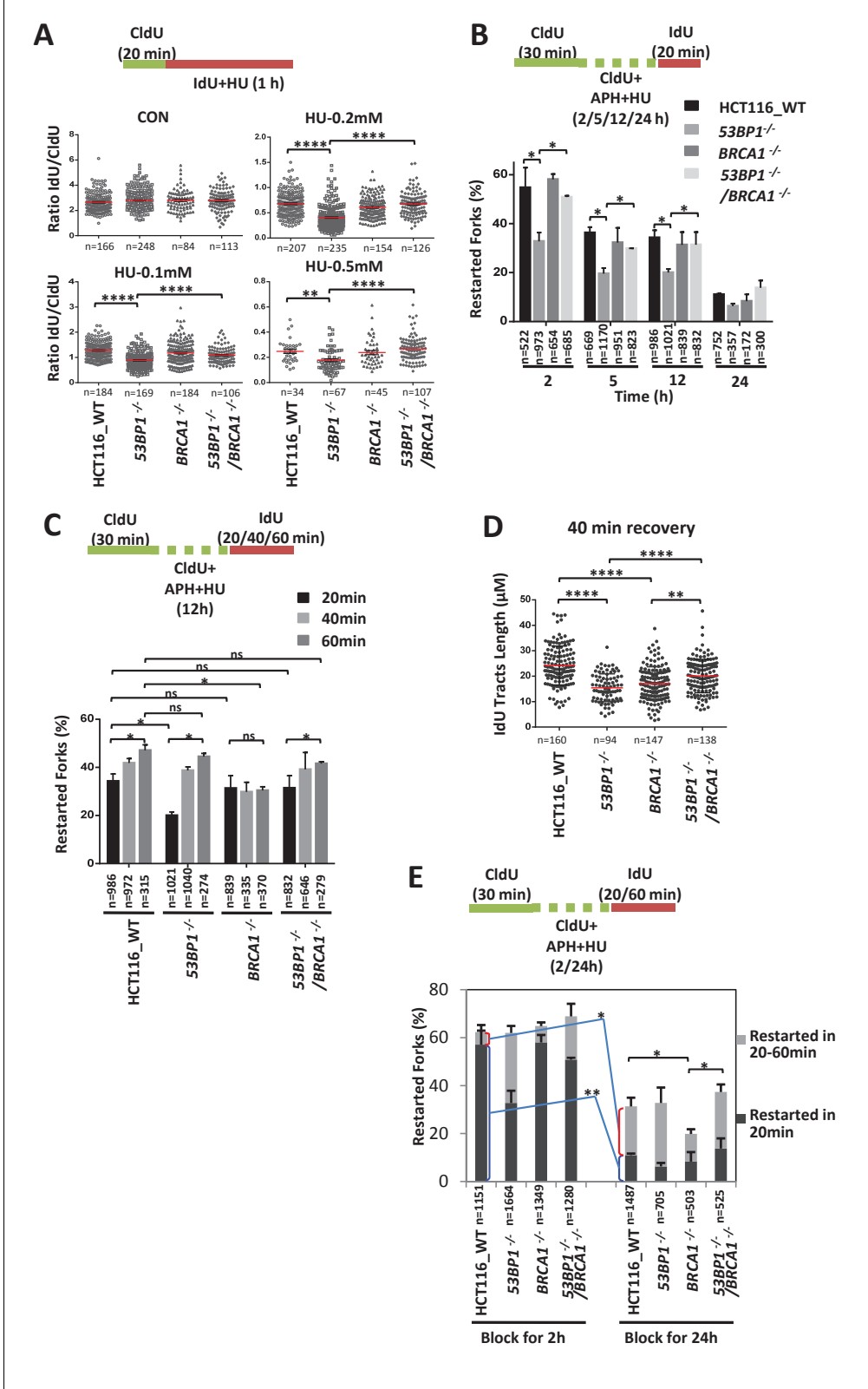

**Figure 3.** 53BP1 and BRCA1 antagonistically promote two distinct fork restart pathways. (**A**) DNA combing assay showing the replication progression under low concentration of HU. The sketch above delineates the experimental design. HCT116 cells were pulse-labeled with CldU for 20 min and then incubated with IdU and HU for 1 hr. The ratios of IdU track compared to CldU track were plotted. The mean and s.e.m are shown. (**B, C**) graphs showing stalled replication fork restart rates after different periods of replication inhibition followed different recovery times. To completely block the

*Figure 3 continued on next page*

Figure 3 continued

replication fork, 5 µM APH and 5 mM HU were added. The mean and s.d. from three independent experiments are shown. (D) A graph showing IdU tracts length after 40 min recovery following 12 hr blocking in (C). (E) A graph showing stalled replication fork restart rates after different periods of replication inhibition followed different recovery times. ****p<0.0001, **p<0.01, *p<0.05, ns p>0.05. Please refer to *Figure 3—figure supplement 1* for additional information in support of *Figure 3*.

DOI: https://doi.org/10.7554/eLife.30523.006

The following figure supplement is available for figure 3:

**Figure supplement 1.** BRCA1 M1775R retains its function in stalled fork restart.

DOI: https://doi.org/10.7554/eLife.30523.007

time (11% at 24 hr; *Figure 3E*), suggesting that this 53BP1-dependent pathway mainly works in the early stage of replication stress. In contrast, the slow-kinetics fork restart pathway (recovered between 20 and 60 min) of wild-type HCT116 cells was not efficient in the early stage (5% at 2 hr; *Figure 3E*), but significantly increased in the late stage (20% at 24 hr) of replication stress, suggesting that this BRCA1-dependent pathway mainly works in the late stage of replication stress.

## 53BP1 and BRCA1 have antagonistic functions in stalled fork cleavage

BRCA1 promotes fork cleavage/unhooking at the initiation step of the Fanconi anemia pathway, in which replication forks are blocked by DNA interstrand crosslinks (*Bunting et al., 2012*; *Long et al., 2014*). These findings prompted us to examine whether BRCA1 and 53BP1 have functions in the cleavage and/or stabilization of stalled forks. We measured the DSB accumulation using a neutral comet assay. We found that more DSBs accumulated in $53BP1^{-/-}$ cells than in wild-type cells after HU treatment (*Figure 4A*). This phenotype was rescued by a full length 53BP1 (*Figure 4—figure supplement 1A and B*), suggesting that it's not caused by off-targeting or clone variation. DSB accumulation may result from increased fork cleavage or impaired DSB repair after fork broken. To distinct these two reasons, we examined DSB accumulation after Olaparib or CPT treatment. Different from HU, these two drugs induce accumulation of broken single-strand DNA, which are then directly converted to broken forks and one-end DSBs during replication without cleavage step. $53BP1^{-/-}$ cells showed similar level of DSBs as that of wild-type cells after Olaparib or CPT treatment (*Figure 4B*), demonstrating that 53BP1 is dispensable for DSB repair of broken forks. Thus, 53BP1 protects stalled forks from breakage (it's distinct from the fork protection from nascent DNA degradation; see Discussion below) but does not promote DSB repair after HU treatment. In contrast, $BRCA1^{-/-}$ cells showed more DSBs than did wild-type cells during Olaparib or CPT treatment (*Figure 4B*), consistent with previous finding that BRCA1 is required for one-end DSB repair through HR (*Bunting et al., 2010*). Surprisingly, in contrast with Olaparib or CPT treatments, $BRCA1^{-/-}$ cells showed fewer DSBs than did wild-type cells after HU treatment (*Figure 4A*). Complementation experiments showed that a full length BRCA1 rescued this defect of $BRCA1^{-/-}$ cells (*Figure 4—figure supplement 1C–E*), suggesting that this phenotype is not due to off-target or clone variation. This phenomenon cannot be explained by its function in DSB repair, suggesting that BRCA1 has additional function to promote the breakage of stalled forks after HU treatment. Interestingly, the DSB accumulation in $53BP1^{-/-}/BRCA1^{-/-}$ cells was counteracted to a level similar to that of wild type cells after HU treatment (*Figure 4A*), suggesting that 53BP1 and BRCA1 have antagonistic functions in the stabilization/cleavage of stalled forks. We measured γH2AX and RPA2-pS4/8, which reflect DSB generation, by quantitative image-based cytometry (QIBC) after replication stress (*Buisson et al., 2015*; *Feng et al., 2016*; *Toledo et al., 2013*). $53BP1^{-/-}$ cells and $BRCA1^{-/-}$ cells showed significantly higher and lower γH2AX and RPA2-pS4/8 signals, respectively, than wild-type cells after HU treatment (*Figure 4C*); while these phenotypes were counteracted in the double knockout cells, consistent with the results of the comet assay. These results suggest that the two proteins had the opposite effect on fork cleavage.

We then tested DT40 knockout cells, which showed similar results (*Figure 4—figure supplement 2A*), suggesting that the antagonism between 53BP1 and BRCA1 in fork breakage is conserved in vertebrates.

Moreover, we examined γH2AX foci formation over a time course after HU treatment (*Figure 4D and E*). In wild-type cells, the γH2AX foci were few in the early stage after HU treatment, but increased progressively over time, consistent with previous results in U2OS cells (*Petermann et al.,*

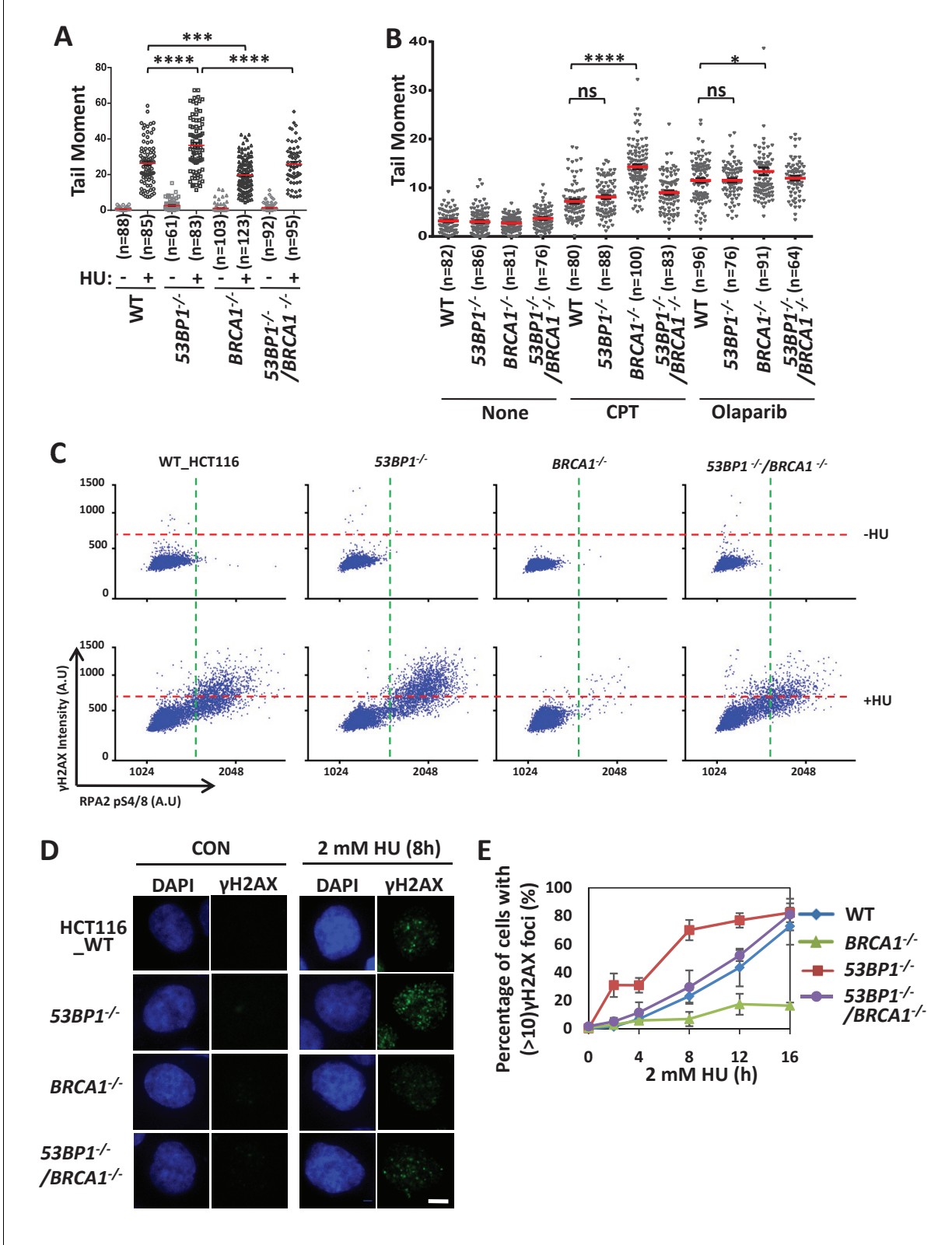

**Figure 4.** 53BP1 and BRCA1 have opposite effects in inducing cleavage of replication forks. (**A, B**) Comet assays measuring DSB accumulation in wild-type, *53BP1*−/−, *BRCA1*−/− and *53BP1*−/− *BRCA1*−/− HCT116 cells treated with HU (2 mM for 12 hr; **A**), CPT (1 μM for 8 hr) and Olaparib (1 μM for 8 hr; **B**). The mean and s.e.m. are shown. ****p<0.0001, ***p<0.001, *p<0.05, ns p>0.05. (**C**) QIBC analysis of immunolabeled wild-type, *53BP1*−/−, *BRCA1*−/− and *53BP1*−/− *BRCA1*−/− HCT116 cells. Asynchronous cells were treated with HU (2 mM) for 5 hr before fixing and immunostaining for γH2AX and RPA2-pS4/8.
*Figure 4 continued on next page*

*Figure 4 continued*

The mean nuclear intensities for γH2AX and RPA2-pS4/8 were determined for each of 5000 individual cells and were plotted. (D, E) Immunostaining (D) and quantification (E) showing γH2AX foci formation over time. The mean and s.d. from three independent experiments are shown. Please refer to *Figure 4—figure supplement 1* and *Figure 4—figure supplement 2* for additional information in support of *Figure 4*.

DOI: https://doi.org/10.7554/eLife.30523.008

The following figure supplements are available for figure 4:

**Figure supplement 1.** Characterization of BRCA1 and 53BP1 knockout HCT116 cells.

DOI: https://doi.org/10.7554/eLife.30523.009

**Figure supplement 2.** 53BP1 and BRCA1 have opposite effects in inducing cleavage of replication forks.

DOI: https://doi.org/10.7554/eLife.30523.010

*2010*). When 53BP1 was disrupted, the γH2AX foci formed more quickly in the early stage, but the level was similar to that in wild-type cells after long-term (16 hr) treatment, suggesting that 53BP1 acts predominantly during short-term replication stress to stabilize stalled forks. In $BRCA1^{-/-}$ cells, the γH2AX foci were similar to wild-type cells in the early stage, but became significantly lower in the later stage (12–16 hr), suggesting that BRCA1 gradually becomes predominant to promote fork breakage over time. Moreover, the double knockout cells showed kinetics of γH2AX foci formation similar to that of wild-type cells, suggesting that the counteraction between 53BP1 and BRCA1 occurs throughout replication stress. These findings are consistent with the fork restart analysis showing that these two proteins counteract each other to promote the early- and late-acting pathways, respectively.

## PTIP, REV7, and CtIP are dispensable for stalled fork stabilization or breakage

Several proteins function downstream of 53BP1 or BRCA1 to regulate the pathway choice of DSB repair, including RIF1 (*Chapman et al., 2013*; *Di Virgilio et al., 2013*; *Escribano-Díaz et al., 2013*; *Feng et al., 2013*; *Zimmermann et al., 2013*), REV7 (*Boersma et al., 2015*; *Xu et al., 2015*) and PTIP (*Callen et al., 2013*; *Wang et al., 2014*) (downstream of 53BP1), and CtIP (downstream of BRCA1)(*Bunting et al., 2010*). We examined whether these proteins also participate in the antagonistic functions of 53BP1 and BRCA1 in promoting the stabilization/cleavage of stalled replication forks. The depletion of RIF1 promoted γH2AX signals and DSB generation after HU treatment (*Figure 4—figure supplement 2B–E*), suggesting that RIF1 participates in the protection of stalled forks, consistent with the finding that RIF1 is required for cells to resist replication stress together with 53BP1. However, the depletion of PTIP and REV7 individually did not affect γH2AX signals and DSB generation under replication stress (*Figure 4—figure supplement 2B–D*), suggesting that unlike in DSB repair, PTIP and REV7 are dispensable for the function of 53BP1 in stalled fork protection. Unlike BRCA1, the depletion of CtIP had no or very weak effect on γH2AX signals and DSB generation (*Figure 4—figure supplement 2B–D*). Consistent with the genetic analysis above, these results suggest that the mechanisms by which 53BP1 and BRCA1 act in the stabilization/cleavage of stalled replication forks are distinct from these in DSB repair.

## BRCA1 promotes the recruitment of the SLX-MUS endonuclease complex to chromosome under replication stress

Several studies have shown that MUS81 mediates the breakage of stalled forks and promotes a late-acting restart pathway (*Franchitto et al., 2008*; *Hanada et al., 2007*; *Pepe and West, 2014*), which is similar to the functions of BRCA1. SLX4, which along with SLX1, MUS81 and EME1 forms a large endonuclease complex (SLX-MUS) (*Fekairi et al., 2009*; *Svendsen et al., 2009*), also takes part in this process under certain conditions (*Couch et al., 2013*; *Ragland et al., 2013*). Therefore, we investigated the relationship between BRCA1 and the SLX-MUS complex. FLAG-immunoprecipitations with cell extracts expressing FLAG-tagged proteins revealed that BRCA1 and SLX4 are mutually present in their immunoprecipitates (*Figure 5A and B*), demonstrating that BRCA1 interacts with SLX4. This interaction was not affected by removing DNA (*Figure 5—figure supplement 1A*), indicating that their association is not mediated by DNA. BRCA1 was observed in the

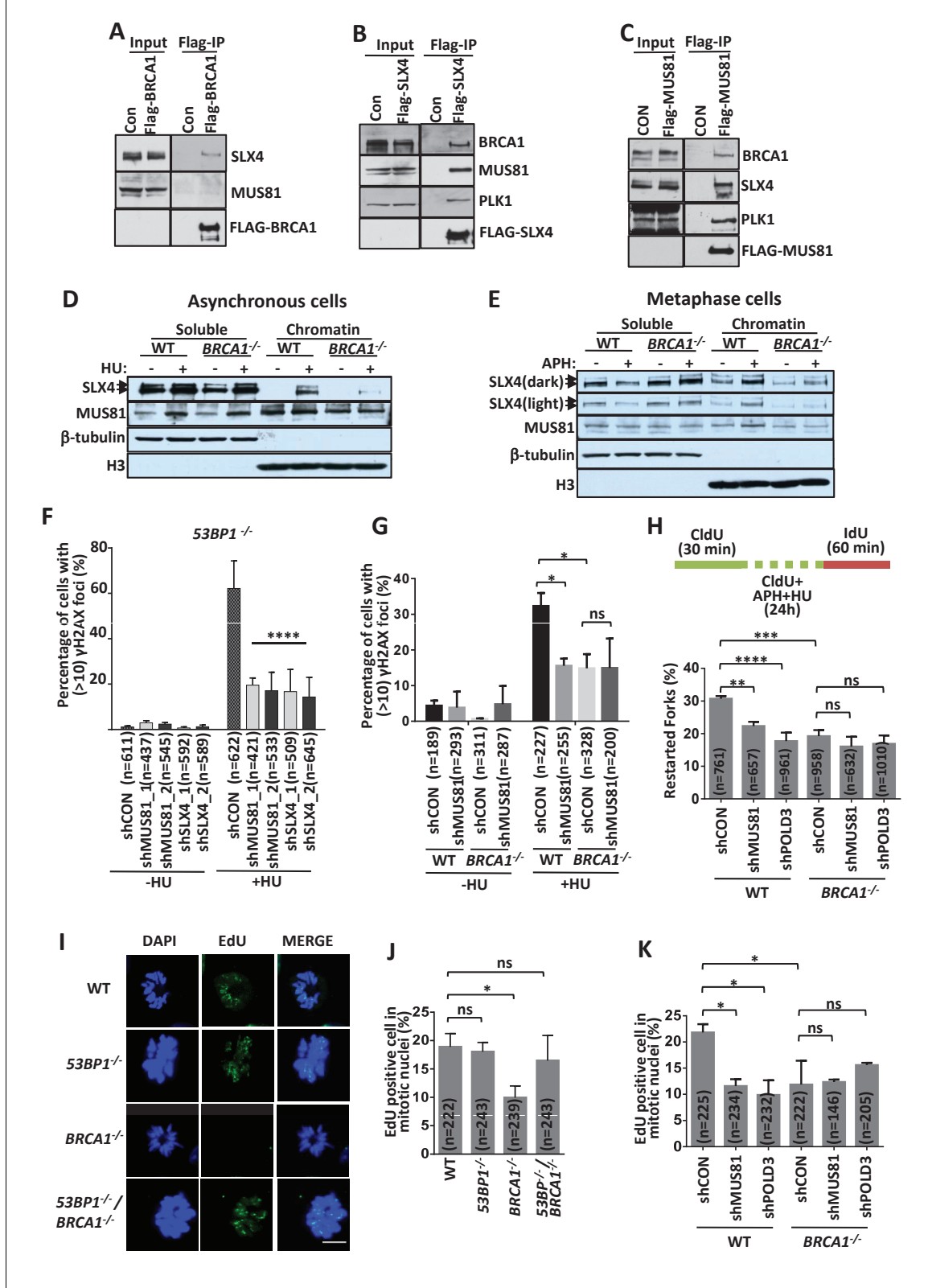

**Figure 5.** BRCA1 promotes SLX-MUS-coupled BIR pathway. (**A–C**) Immunoblot showing the immunoprecipitation from the extracts of HEK293 cells transfected with vectors expressing Flag-tagged BRCA1 (**A**), SLX4 (**B**), MUS81 (**C**) or control vector (Con). (**D, E**) Immunoblot showing protein level of SLX4 and MUS81 on the chromosome. Asynchronous cells were treated with or without 2 mM HU for 16 hr (**D**). Metaphase cells (**E**) were prepared as experimental workflow in *Figure 5—figure supplement 1B*. (**F, G**) Graphs showing γH2AX foci formation in MUS81- or SLX4-depleted *53BP1⁻/⁻* (**F**) cells

*Figure 5 continued on next page*

Figure 5 continued

and *BRCA1*[-/-] (G) HCT116 cells treated with or without 2 mM HU for 3 hr. Knockdown efficiency was showed in **Figure 5—figure supplement 1F and G**. (H) A graph showing stalled replication fork restart rates in MUS81- or POLD3- depleted *BRCA1*[-/-] cells. (I, J) Immunofluorescence (I) and its quantifications (J) showing DNA synthesis (EdU foci, green) in condensed mitotic nuclei (DAPI, blue). Experiments were preformed as the workflow in **Figure 5—figure supplement 1C**. (K) A graph showing mitotic DNA synthesis in MUS81- or POLD3- depleted *BRCA1*[-/-] cells. The mean and s.d. from three independent experiments are shown. ****p<0.0001, ***p<0.001,**p<0.01, *p<0.05, ns p>0.05.
DOI: https://doi.org/10.7554/eLife.30523.011
The following figure supplement is available for figure 5:

**Figure supplement 1.** BRCA1 promotes the recruitment of the SLX-MUS complex to stalled forks.
DOI: https://doi.org/10.7554/eLife.30523.012

immunoprecipitate of FLAG-MUS81, but FLAG-BRCA1 did not pulldown a significant band of MUS81 (*Figure 5A and C*), implying that the interaction of MUS81 with BRCA1 may be weak.

We then examined whether BRCA1 is required for the recruitment of the SLX-MUS complex to stalled replication forks. SLX4 accumulated in the chromatin fraction after HU treatment (*Figure 5D*) and this recruitment was significantly reduced in *BRCA1*[-/-] cells, suggesting that BRCA1 may promote the recruitment of SLX4 to stalled forks. The MUS81 levels in the chromatin fraction did not change significantly after HU treatment and were also not affected by BRCA1 deficiency (*Figure 5D*), suggesting that the majority of MUS81 is SLX4-free and recruited to the chromatin independent of fork stalling and BRCA1. These results are consistent with a previous study that the interaction of MUS81 with SLX4 is weak in asynchronous cells (*Matos et al., 2011*; *Wyatt et al., 2013*).

The SLX-MUS complex is recruited to mitotic chromatin and promotes mitotic DNA synthesis (MiDAS) (*Bhowmick et al., 2016*; *Minocherhomji et al., 2015*). We examined whether this recruitment is also dependent on BRCA1. Both SLX4 and MUS81 levels on mitotic chromatin are increased after APH treatment in the wild-type cells, but not in *BRCA1*[-/-] cells (*Figure 5E* and *Figure 5—figure supplement 1B*). Moreover, immunofluorescence experiments showed that the recruitment of MUS81 to MiDAS sites was significantly decreased in *BRCA1*[-/-] cells, but weakly increased in *53BP1*[-/-] cells, compared to that in wild-type cells (*Figure 5—figure supplement 1C–E*). These results suggest that BRCA1 promotes the recruitment of the SLX-MUS complex to mitotic chromatin on MiDAS sites, while 53BP1 suppresses it.

## BRCA1 promotes the MUS81-coupled BIR pathway

We then examined whether SLX-MUS has a similar function as BRCA1 in counteracting 53BP1 in fork stabilization. As shown in *Figure 5F*, the depletion of the SLX-MUS complex components MUS81 and SLX4 strongly suppressed stalled fork breakage in *53BP1*[-/-] cells, similar to the effects of BRCA1 disruption. Moreover, MUS81-depleted wild-type and *BRCA1*[-/-] cells resulted in a similar suppression of fork cleavage (*Figure 5G*), suggesting that BRCA1 and the SLX-MUS complex are in the same fork cleavage pathway. Together, these results suggest that BRCA1 promotes fork cleavage through recruiting the SLX-MUS endonuclease complex.

MUS81 and SLX4 are required for the late-acting fork restart pathway and MiDAS via a BIR-like mechanism (*Hanada et al., 2007*; *Minocherhomji et al., 2015*). We therefore examined whether BRCA1 has a similar function in BIR. Epistasis analysis through the determination of the late-acting fork restart pathway revealed that BRCA1 is in the same pathway as MUS81 and POLD3 (*Figure 5H*), which play a key role in BIR (*Bhowmick et al., 2016*; *Costantino et al., 2014*; *Sotiriou et al., 2016*). Thus, the BRCA1-dependent pathway is a cleavage-coupled BIR.

## Mitotic replication restart specifically requires the BRCA1-mediated pathway

We also examined whether BRCA1 and 53BP1 play roles in MiDAS. Wild-type and mutant cells were treated with replication stress in the form of a low dose (0.2 μM) of APH and then 5-ethynyl-2'deoxy-uridine (EdU) was added for 30 min to visualize new DNA synthesis (*Figure 5—figure supplement 1C*) as previously described (*Minocherhomji et al., 2015*). Approximately 18% of the mitotic wild-type HCT116 cells contained EdU foci (*Figure 5I and J*), as previously reported (*Minocherhomji et al., 2015*). EdU-positive mitotic cells were significantly decreased in *BRCA1*[-/-]

cells (approximately 9%), but not in *53BP1⁻/⁻* cells (approximately 17%), demonstrating that BRCA1, but not 53BP1, is specifically required for MiDAS. Moreover, *53BP1⁻/⁻/BRCA1⁻/⁻* cells showed more EdU-positive mitotic cells (17%) than *BRCA1⁻/⁻* cells (*Figure 5I and J*), suggesting that 53BP1 also has a BRCA1-antagonistic function in MiDAS.

Moreover, epistasis analysis showed that BRCA1 was also in the same pathway as MUS81 and POLD3 in MiDAS (*Figure 5K*), suggesting that BRCA1 also promotes BIR in MiDAS.

## PLK1 expression is increased during replication stress and mitosis

Our data revealed that the 53BP1- and BRCA1-dependent pathways occur predominantly in the early and late (particularly in mitosis) stages of replication stress, respectively. A striking question is how cells temporally regulate the switch between these two pathways. PLK1 promotes stalled fork breakage in the presence of an ATR inhibitor (*Ragland et al., 2013*), inhibit 53BP1 function (*Lee et al., 2014*; *Orthwein et al., 2014*) and stimulate the assembly and activity of the SLX-MUS complex during mitosis (*Matos et al., 2011*; *Wyatt et al., 2013*). The expression level of PLK1 is regulated during the cell cycle and peaks at the M phase (*Barr et al., 2004*). Importantly, our ongoing interactome analysis of DNA repair proteins revealed that PLK1 interacts with BRCA1, SLX4 and 53BP1 (*Figures 5B,C and* and *6A*). We therefore speculated that increased PLK1 activity might control the conversion of these two restart pathways when replication stress is prolonged. We first examined the PLK1 levels after replication inhibition (*Figure 6B and C*). The cells were arrested in G1/S phase using double-thymidine block and were then treated with HU after release into S phase for 2 hr. The PLK1 levels increased to more than 2-fold after 8–12 hr of HU treatment. We also examined PLK1 levels in unperturbed cells using a QIBC assay (*Figure 6D* and *Figure 6—figure supplement 1A*). The cells were pulse-labeled with EdU 30 min before HU treatment and then pre-extracted before fixing and staining. The PLK1 levels were highest in the G2/M phase without HU treatment. After 8–12 hr of HU treatment, the G2/M phase cells were shifted to the G1 and/or G1/S phases, whereas the S phase cells were blocked (*Figure 6—figure supplement 1A*). Interestingly, the PLK1 levels in the S phase were significantly increased after HU treatment over time (*Figure 6D* and *Figure 6—figure supplement 1A*). These results suggest that prolonged replication stress induces PLK1 expression in S phase cells.

## PLK1 regulates the stalled replication restart pathways

We then examined whether PLK1 regulates the replication restart pathways. We checked the assembly of the SLX-MUS complex (*Figure 6E*). The associated levels of SLX4 were significantly increased when FLAG-MUS81 was immunoprecipitated from mitotic extracts, in which PLK1 was highly expressed. This interaction was dramatically decreased by PLK1 inhibitors (*Figure 6E* and *Figure 6—figure supplement 1B*), suggesting that PLK1 activity promotes the assembly of the SLX-MUS complex, consistent with previous studies (*Matos et al., 2011*; *Wyatt et al., 2013*). Consistently, the recruitment of MUS81 to chromatin and stalled forks was impaired by PLK1 inhibitor (*Figure 6—figure supplement 1C–E*). In contrast, the interaction of BRCA1 with SLX4 and their recruitment to stalled forks were not impaired by PLK1 inhibitors (*Figure 6—figure supplement 1B,C,F and G*).

Then, we tested SLX-MUS complex-mediated fork breakage when the PLK1 inhibitor was present (*Figure 6F*). The PLK1 inhibitor strongly suppressed replication stress-induced fork breakage in both wild-type and *53BP1⁻/⁻* cells. Moreover, we examined whether PLK1 activity is required for the slow-kinetics fork restart pathway. The PLK1 inhibitor strongly suppressed the slow-kinetics fork restart pathway in the wild-type cells, but not in the *BRCA1⁻/⁻* cells (*Figure 6G*), demonstrating that PLK1 promotes the BRCA1-dependent fork cleavage pathway.

## Discussion

### 53BP1-RIF1 and BRCA1 have new functions in the restart of stalled replication forks

Here, we showed that 53BP1 has a new function in the cleavage-free fork restart pathway, which protects forks from breakage, has fast-kinetics and mainly works in the early stage of replication stress (*Figure 7*). This pathway has been widely described and many DNA remodeling enzymes, including helicases, translocases, and recombinases (*Cortez, 2015*; *Neelsen and Lopes, 2015*),

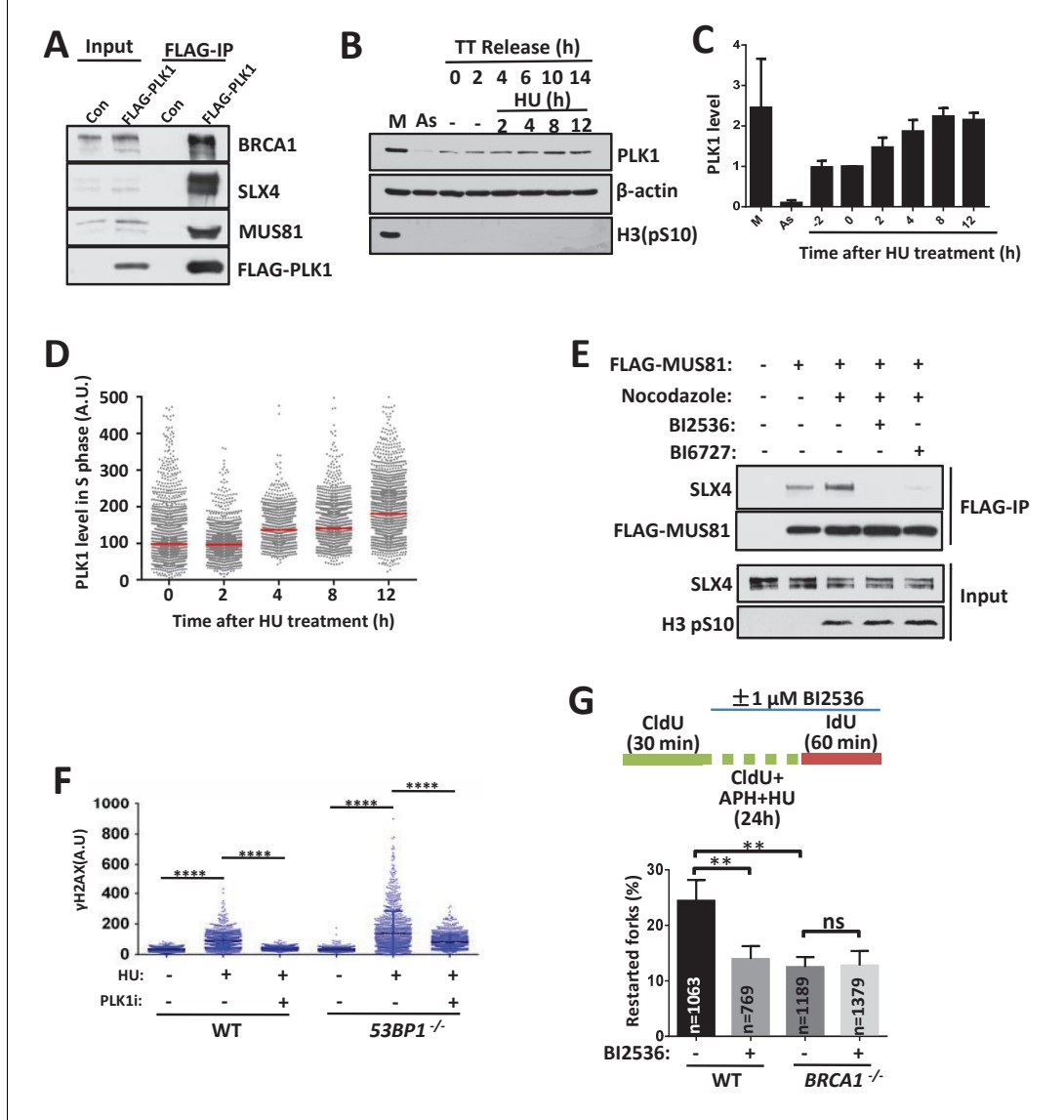

**Figure 6.** PLK1 controls the switching between the 53BP1-mediated cleavage free pathway to BRCA1-mediated cleavage pathway. (A) Immunoblot showing the immunoprecipitation from extracts of HEK293 cells tranfected with control vector (Con) or vector expressing Flag-tagged PLK1. (B, C) Immunoblotting (B) and its quantification (C) showing PLK1 levels after replication inhibition. The mean and s.d. from three independent experiments are shown. (D) QIBC analysis of PLK1 expression after replication inhibition. Asynchronous HCT116 cells were treated with 2 mM HU as indicated time before fixing. PLK1 levels of S phase (*Figure 6—figure supplement 1A*) were gated and plotted. Red lines indicate the medians of PLK1. (E) Immunoblot showing the immunoprecipitation of FLAG-tagged MUS81. Suspension HEK293 cells expressing FLAG-tagged MUS81 were treated with or without nocodazole (100 ng/ml) and PLK1 inhibitors (10 μM BI2536 or BI6727) for 17 hr and 5 hr before harvest, respectively. (F) QIBC analysis of wild-type and *53BP1*$^{-/-}$ HCT116 cells treated with 2 mM HU and PLK1 inhibitor (10 μM BI2536) for 3 hr. ****p<0.0001. (G) DNA combing assay showing that PLK1 works in the same pathway with BRCA1 in stalled fork restart. The sketch above delineates the experimental design. The mean and s.d. from three independent experiments are shown. **p<0.01, ns p>0.05. Please refer to *Figure 6—figure supplement 1* for additional information in support of *Figure 6*.

DOI: https://doi.org/10.7554/eLife.30523.013

The following figure supplement is available for figure 6:

**Figure supplement 1.** PLK1 promotes the recruitment of MUS81 to stalled replication forks.

DOI: https://doi.org/10.7554/eLife.30523.014

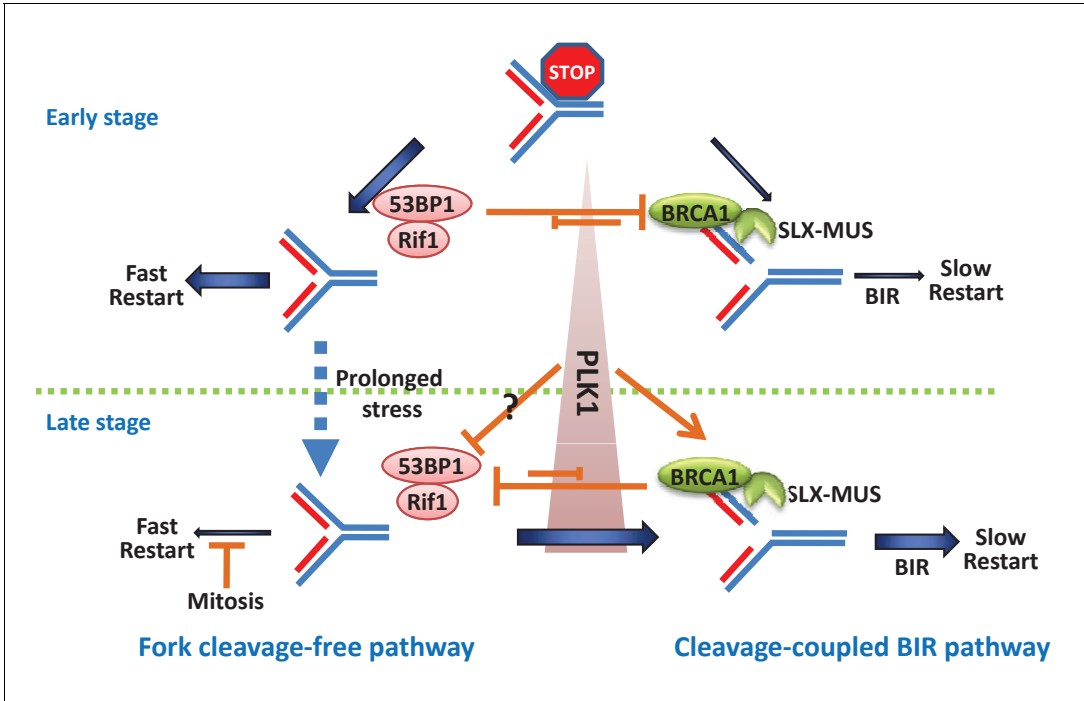

**Figure 7.** A model showing the pathway choice of stalled replication restart by 53BP1 and BRCA1.
DOI: https://doi.org/10.7554/eLife.30523.015

participate in it. RIF1, the downstream protein of 53BP1 in NHEJ repair, is also in the same pathway with 53BP1 in response to replication stress. RIF1 has been reported to restart stalled forks along with BLM, which is a fork remodeling enzyme (*Davies et al., 2007*; *Xu et al., 2010*). Therefore, mechanistically, 53BP1-RIF1 may protect and repair stalled forks through BLM-mediated fork remodeling. It remains to be studied in future how 53BP1 and RIF1 exactly protect stalled replication forks.

Conversely, BRCA1 has a new function in the cleavage-coupled BIR pathway, mechanistically through promoting the recruitment of the MUS-SLX complex to stalled forks (*Figure 7*). This pathway has slow-kinetics and predominantly acts after long periods of replication inhibition, especially during mitosis. MiDAS is possibly a special situation of replication restart, in which forks were persistently stalled until detected during chromosome condensation in mitosis and repaired specifically by the cleavage-coupled BIR pathway.

There are two types of fork protections: one is to protect the stalled forks from cleavage by SLX-MUS to generate one-end DSB, as discussed here; and the other is to protect stalled forks from resection by MRE11 to degrade nascent DNA as described by Schlacher, K. *et.al* (*Schlacher et al., 2011*; *Schlacher et al., 2012*). BRCA1 has opposite activities in these two processes: it promotes fork breakage by SLX-MUS, but inhibits fork degradation by MRE11 (*Schlacher et al., 2012*). These two functions of BRCA1 likely have no directly causality because of two reasons: first, the defect of nascent DNA protection in the BRCA1 mutant cannot be rescued by disrupting 53BP1 (*Ray Chaudhuri et al., 2016*), which is different from their antagonistic functions in fork restart described here; second, BRCA2, which plays an essential role in the nascent DNA protection (*Schlacher et al., 2011*), is dispensable for BIR and MiDAS (*Bhowmick et al., 2016*; *Feng and Jasin, 2017*; *Lai et al., 2017*; *Sotiriou et al., 2016*), suggesting that these functions are two independent events. In fact, BRCA2 has opposite function from BRCA1 to protect stalled forks from breakage (*Lomonosov et al., 2003*). Moreover, BRCA2-defective cells display delayed restart of stalled forks (*Ray Chaudhuri et al., 2016*) and this delayed restart is dependent on MUS81 (*Lai et al., 2017*; *Lemaçon et al., 2017*), suggesting that BRCA2 may work in a MUS81-independent pathway to restart stalled forks. Thus, it will be interesting to study whether BRCA2 functions in the cleavage-free replication restart pathway together with 53BP1 in future.

## 53BP1 and BRCA1 counteract each other to control the time-dependent switch of the fork restart pathways

In the early stage of replication stress, stalled forks were restarted efficiently and quickly (*Figure 3B*), suggesting that most stalled forks kept activation at an early stage. The stalled forks were not more restarted by the fast pathway when stress was prolonged, even without BRCA1-dependent cleavage (*Figure 3B*; $BRCA1^{-/-}$ cells showed a similar restart rate as the wild-type cells after 24 hr of inhibition followed by a 20 min recovery), suggesting that these forks might become inactivated or collapsed, such as via replisome-dissociation or over-regression into an aberrant Holiday junction. In contrast, stalled forks in 53BP1-deficient cells were broken by BRCA1-dependent cleavage in the early stress stage (*Figure 4D and E*) and then were restarted by a slow pathway, BIR (*Figure 3C and E*). When BRCA1 was disrupted, this cleavage in 53BP1-deficient cells was suppressed (*Figure 4D and E*) and the stalled forks could still be recovered by the fast restart pathway (*Figure 3B*; $BRCA1^{-/-}53BP1^{-/-}$ cells showed a higher fork restart rate than $53BP1^{-/-}$ cells), suggesting that the stalled forks were not inactivated/collapsed before cleavage in the early stage. Thus, the BIR pathway not only repairs broken/collapsed forks, but also restarts stalled forks by coupling BRCA1 and MUS-SLX complex-mediated cleavage. Moreover, fork cleavage and the cleavage-coupled BIR pathway in BRCA1-deficient cells were suppressed by 53BP1 in the late stage or during mitosis (*Figures 3C,E*, *4D,E and* and *5I,J*; $BRCA1^{-/-}53BP1^{-/-}$ cells showed a higher fork cleavage efficiency and fork restart rate than $BRCA1^{-/-}$ cells), suggesting that 53BP1 can also protect inactivated/collapsed forks. Together, 53BP1 and BRCA1 counteract each other to protect/cleave damaged (stalled and collapsed) forks and promote replication restart through two distinct pathways.

There is a balance between these two mutually exclusive fork restart pathways. In the early stage of replication stress, the balance favors to 53BP1-dependent pathway, although the BRCA1-dependent pathway is also operative. When replication stress is prolonged, the balance tilts toward the BRCA1-dependent pathway. The switch from 53BP1-mediated pathway to BRCA1-mediated pathway is likely achieved by the cell-cycle-dependent regulation of PLK1 activity, which is high in mitotic cells or in S-phase cells upon prolonged HU treatment (*Figure 7*). PLK1 promotes the cleavage pathway by enhancing the interactions within the SLX-MUS complex. Moreover, PLK1 can also inactivate 53BP1, at least during mitosis (*Lee et al., 2014*; *Orthwein et al., 2014*), but it is unclear whether this inactivation occurs in the S phase when replication stress is prolonged. These findings indicate that fork-breakage/cleavage is not the passive consequence of collapse but a programmed process, combining the temporal regulations of the assembly of the MUS-SLX complex, the increase in PLK1 activity, and the de-repression of 53BP1-RIF1 by BRCA1.

## The antagonistic interactions between 53BP1 and BRCA1 in replication restart are independent of their roles in DSB repair

Our data showed that the functions of 53BP1-RIF1 and BRCA1 in replication restart are clearly different from their roles in DSB repair. First, only the defect in fork restart but not DSB repair of the $53BP1^{-/-}$ cells was rescued by the disruption of BRCA1. Second, more importantly, we found function-separated mutant of BRCA1. Consistent with this finding, an ubiquitin ligase-inactive mutant of BARD1, the partner of BRCA1, also showed separated functions: it couldn't repair DSBs, but retains its role in response to replication stress (*Densham et al., 2016*).

Moreover, the antagonistic functions of 53BP1 and BRCA1 in replication restart also mimic their counteracting functions in DSB repair in some ways. In both processes, 53BP1 and BRCA1 mutually counteracts at initiation steps. The decision step of DSB repair pathway choice is the end resection, which is initiated by the CtIP-MRN endonuclease complex. Similarly, the decision step of fork restart pathway conversion is the fork cleavage, which is performed by the MUS-SLX endonuclease complex. It is possible that 53BP1 and RIF1 might have a common mechanism in both blocking DSB resection and preventing fork cleavage, such as forming a higher-order chromatin structure through their oligomerization domains to suppress the access of BRCA1-recruited nucleases as speculated previously (*Panier and Boulton, 2014*). This chromatin access-limiting function is not mutually exclusive with their potential ability to recruit the downstream proteins, such as BLM. Conversely, BRCA1-recuited nucleases might generate products unsuitable for 53BP1 and RIF1-binding. Moreover, BRCA1 might destabilize the chromatin structures that are necessary for 53BP1 and RIF1

accumulation. In support of this notion, BRCA1 has been shown to have a chromatin-decondensation activity (*Ye et al., 2001*).

Oncogene activation-induced replication stress is common in cancer cells, and stalled replication forks are a major cause of genome instability in tumorigenesis (*Hills and Diffley, 2014*). Moreover, DNA replication is one of the most common drug targets for cancer therapy. Insights into the pathways selected by cells to counteract replication stress may provide new drug targets and could also be exploited to modulate therapeutic responses in a clinically relevant manner.

# Materials and methods

## Key resources table

| Reagent type (species) or resource | Designation | Source or reference | Identifiers | Additional information |
|---|---|---|---|---|
| genetic reagent (*Homo sapiens*) | MUS81 (shRNA) | Sigma-Aldrich (St. Louis, MO, USA) | TRCN0000049726, TRCN0000049727 | |
| genetic reagent (*Homo sapiens*) | SLX4 (shRNA) | Sigma-Aldrich (St. Louis, MO, USA) | TRCN0000143727, TRCN0000142519 | |
| genetic reagent (*Homo sapiens*) | POLD3 (shRNA) | Sigma-Aldrich (St. Louis, MO, USA) | TRCN0000052990 | |
| genetic reagent (*Homo sapiens*) | BRCA1 (siRNA) | PMID:16109739 | | |
| genetic reagent (*Homo sapiens*) | RIF1 (siRNA) | PMID:20711169 | | |
| genetic reagent (*Homo sapiens*) | CtIP (siRNA) | PMID:23333306 | | |
| genetic reagent (*Homo sapiens*) | PTIP (siRNA) | PMID:15456759 | | |
| genetic reagent (*Homo sapiens*) | REV7 (siRNA) | PMID:23287467 | | |
| cell line (*Homo sapiens*) | HCT116 | ATCC | CCL-247 | |
| cell line (*Homo sapiens*) | Hela | ATCC | CCL-2 | |
| cell line (*Homo sapiens*) | 293T | ATCC | CRL-3216 | |
| cell line (*Homo sapiens*) | HEK293 Suspension | ATCC | CRL-1573.3 | |
| cell line (*Gallus gallus*) | DT40 | other | | A gift from Dr. Minoru Takata's lab |
| antibody | anti-BrdU (BU1/75) (mouse monoclonal) | BD Biosciences (San Jose, CA, USA) | 347580 | IF: 1:250 |
| antibody | anti-BrdU (B44) (rat monoclonal) | Abcam (Cambridge, UK) | ab6326 | IF:1:50 |
| antibody | Flag (mouse monoclonal) | MBL (Japan) | M185-3L | WB:1:2000 |
| antibody | γH2AX (mouse monoclonal) | Millipore (St. Louis, MO, USA) | 05–636 | IF:1:5000 |
| antibody | RPA2 (rabbit polyclonal) | Bethyl (Montgomery, TX, USA) | A300-244A | WB:1:1000; IF:1:500 |
| antibody | MUS81 (rabbit polyclonal) | Proteintech (China) | 11018–1-AP | WB:1:1000 |
| antibody | MUS81 (mouse monoclonal) | Abcam (Cambridge, UK) | ab14387 | IF:1:250 |
| antibody | BRCA1 (rabbit polyclonal) | Millipore (St. Louis, MO, USA) | 07–434 | WB:1:1000 |
| antibody | CtIP (rabbit polyclonal) | Abcam (Cambridge, UK) | ab155988 | WB:1:1000 |

*Continued on next page*

*Continued*

| Reagent type (species) or resource | Designation | Source or reference | Identifiers | Additional information |
|---|---|---|---|---|
| antibody | PTIP (rabbit polyclonal) | Abcam (Cambridge, UK) | ab70434 | WB:1:1000 |
| antibody | REV7 (mouse monoclonal) | BD Biosciences (San Jose, CA, USA) | 612266 | WB:1:1000 |
| antibody | FANCD2 (rabbit polyclonal) | Homemade | | WB:1:1000;IF:1:250 |
| antibody | β-actin (mouse monoclonal) | MBL (Japan) | M177-3 | WB:1:1000 |
| antibody | PLK1 (mouse monoclonal) | Santa Cruz (Dallas, TX, USA) | F-8 | WB:1:1000;IF:1:250 |
| antibody | PLK1 (rabbit polyclonal) | Proteintech (China) | 10305–1-AP | WB:1:1000 |
| antibody | Histone H3-pS10 (mouse monoclonal) | Cell Signaling (Danvers, MA, USA) | 9706 s | WB:1:1000 |
| antibody | Histone H3 (rabbit polyclonal) | Novus Biologicals (Littleton, USA) | NB500-171 | WB:1:1000 |
| antibody | 53BP1 (mouse monoclonal) | Millipore (St. Louis, MO, USA) | MAB3802 | WB:1:1000 |
| antibody | BARD1 (rabbit polyclonal) | proteintech (China) | ab22964-1-AP | WB:1:2000 |
| antibody | Donkey anti-mouse (A594, A488) | Invitrogen (Waltham, Massachusetts, USA) | A21203,A21202 | IF:1:250 |
| antibody | Donkey anti-rabbit (A594) | Invitrogen (Waltham, Massachusetts, USA) | A21207 | IF:1:250 |
| antibody | Donkey anti-rabbit (A488) | Jackson Immunoresearch (Baltimore, MD, USA) | 711-546-152 | IF:1:250 |
| antibody | Donkey anti-rat (A488) | Invitrogen(Waltham, Massachusetts, USA) | A21208 | IF:1:250 |
| antibody | Anti-Mouse IgG | Jackson ImmunoResearch (Baltimore, MD, USA) | 115-035-146, Lot111590 | WB:1:5000 |
| antibody | Anti-Rabbit IgG | Jackson ImmunoResearch (Baltimore, MD, USA) | | WB:1:5000 |
| recombinant DNA reagent | pDEST26-HF (Gateway vector) | this paper | | Progentiors: pDEST26 from Invitrogen |
| recombinant DNA reagent | Flag-SLX4 (plasmid) | this paper | | Progentiors: pDONR221-SLX4; Gateway vector:pDEST36-HF |
| recombinant DNA reagent | Flag-MUS81 (plasmid) | this paper | | Progentiors: pDONR221-MUS81; Gateway vector:pDEST36-HF |
| recombinant DNA reagent | Flag-PLK1 (plasmid) | this paper | | Progentiors: pDONR221-PLK1; Gateway vector:pDEST36-HF |
| recombinant DNA reagent | Flag-BRCA1 (plasmid) | this paper | | Progentiors: pDONR221-BRCA1; Gateway vector:pDEST36-HF |
| commercial assay or kit | comet assay kit | Trivegen (Gaithersburg, USA) | 4250–050 K | |
| chemical compound, drug | HU (hydroxyurea) | Sigma-Aldrich (St. Louis, MO, USA) | V900323 | |
| chemical compound, drug | APH (aphidicolin) | abcam (Cambridge, UK) | ab142400 | |
| chemical compound, drug | ICRF193 | Sigma-Aldrich (St. Louis, MO, USA) | I4659 | |
| chemical compound, drug | Olaparib | selleck (Houston, TX, USA) | S1060 | |
| chemical compound, drug | CPT (Camptothecin) | Sigma-Aldrich (St. Louis, MO, USA) | C9911 | |
| chemical compound, drug | BI2536 | selleck (Houston, TX, USA) | S1109 | |

*Continued on next page*

*Continued*

| Reagent type (species) or resource | Designation | Source or reference | Identifiers | Additional information |
|---|---|---|---|---|
| chemical compound, drug | BI6727 | selleck (Houston, TX, USA) | S2235 | |
| chemical compound, drug | IdU | Sigma-Aldrich (St. Louis, MO, USA) | I7125 | |
| chemical compound, drug | CldU | Sigma-Aldrich (St. Louis, MO, USA) | C6891 | |
| chemical compound, drug | EdU | Sigma-Aldrich (St. Louis, MO, USA) | 900584 | |
| chemical compound, drug | BrdU | Sigma-Aldrich (St. Louis, MO, USA) | B5002 | |
| chemical compound, drug | Thymidine | Sigma-Aldrich (St. Louis, MO, USA) | T1895 | |
| chemical compound, drug | Nocodazole | Sigma-Aldrich (St. Louis, MO, USA) | M1404 | |
| chemical compound, drug | RO3306 | selleck (Houston, TX, USA) | S7747 | |
| chemical compound, drug | XL413 | selleck (Houston, TX, USA) | S7547 | |
| software, algorithm | CellProfiler | Carpenter lab website | | open-source, public domain software |
| software, algorithm | Image J | National Institutes of Health | | public domain, Java |
| software, algorithm | Huygens Professional | Scientific Volume Imaging | | |
| software, algorithm | casplab | open-source, public domain software | | |
| software, algorithm | GraphPad Prism | open-source | | |
| other | DAPI stain | Invitrogen (Waltham, Massachusetts, USA) | | |

## Cell culture and transfection

HeLa cells were cultured in DMEM medium containing 10% fetal bovine serum (FBS; Invitrogen). HCT116 cells were cultured in RPMI1640 medium containing 10% FBS (Invitrogen). HEK293 suspension cells were cultured in Freestyle medium (Invitrogen) supplemented with 1% Gibco FBS and 1% glutamine in an incubator with shaking at 130 r.p.m. DT40 cells were gifted from Dr. Minoru Takata, and grown at 39.5°C, 5% $CO_2$ in RPMI 1640 medium (Gibco) supplemented with 10% fetal calf serum, 1% chicken serum. The other cell lines studied were obtained from the ATCC. All cell lines are not among those listed as commonly misidentified by the International Cell Line Authentication Committee. All cell lines were subjected to mycoplasma testing twice per month and found to be negative. The identity of the cell lines was validated by STR profiling (ATCC) and by analysis of chromosome number in metaphase spreads.

For synchronization, the cells were cultured in medium supplemented with 2.5 mM thymidine for 16 hr and released into fresh medium for 8 hr. The cells were then treated with a second dose of 2.5 mM thymidine for 16 hr and released into fresh medium.

HEK293 suspension cells were transfected with PEI. HeLa and HCT116 cells were transfected with Fugene HD (Promega). The siRNAs targeting Rif1 (5'-GCAGCUUAUGACUACUAAA-3'), CtIP (5'-GCUAAAACAGGAACGAAUC-3'), PTIP (5'-UGCACUAGCCUCACACAUA-3' and 5'-UGUUUGCAAUUGCGGAUUAUU-3') and REV7 (5'-GAUGCAGCUUUACGUGGAA-3'), were transfected using RNAi MAX (Invitrogen). To produce the MUS81 (CCGGGAGTTGGTACTGGATCACATTCTCGAGAATGTGATCCAGTACCAACTCTTTTTG and CCGGCCTAATGGTCACCACTTCTTACTCGAGTAAGAAGTGGTGACCATTAGGTTTTTG), SLX4 (CCGGGCTGGAGCTAGAACAAACCAACTCGAGTTGGTTTGTTCTAGCTCCAGCTTTTTG and CCGGGCTCCTCATCCAGTATGTGAACTCGAGTTCACATACTGGATGAGGAGCTTTTTG), and POL3 (CCGGCGAGTAGCATTATCTGATGATCTCGAGATCA

TCAGATAATGCTACTCGTTTTTG) shRNA, lentiviral plasmids were co-transfected into 293 T cells using PEI. After 4 days, the supernatants containing the packaged lentivirus were harvested and stored at −80°C until further use.

## DT40 cells

The generation of $RIF1^{-/-}$ cells was described as previously by Xu D. *et al* (*Xu et al., 2010*). The generations of $Ku70^{-/-}$, $BRCA1^{-/-}$, $53BP1^{-/-}$, $BRCA1^{-/-}53BP1^{-/-}$ and $RIF1^{-/-}BRCA1^{-/-}$ DT40 cells were described as previously by Escribano-Díaz C. *et al* (*Feng et al., 2013*). The Ku70, 53BP1 and BRCA1 knockout constructs were gifts from Dr. Minoru Takata (*Takata et al., 1998*), Dr. Yoshihito Taniguchi (*Nakamura et al., 2006*) and Dr. Douglas K. Bishop (*Martin et al., 2007*), respectively.

## Generation of BRCA1 and 53BP1-knockout cells

BRCA1- and 53BP1-deficient HCT116 cells were generated using CRISPR. Briefly, guide sequences (BRCA1: CTGAGAAGCGTGCAGCTGAG and GAAGGTAAAGAACCTGCAAC; 53BP1: GCAGCTCTC TGGTCAGAGGT) were inserted into the pX330 vector (*Cong et al., 2013*). The guide-sequence-containing pX330 plasmids were transfected into HCT116 cells and single colonies were picked after 8–10 days of incubation. The genomic fragments of the *BRCA1* and *53BP1*gene were amplified by PCR using the following primers: ctgcttgtgaattttctgagacggatg and GCTCCTTGCTAAGCCAGGCTG TTTG (for BRCA1 site); gtgtcaatctgagaagtgcaactg and CCTAAGACTCTCAGGCACATACTG (for 53BP1). The products were digested with PvuII and PstI, respectively. Colonies containing the expected PCR fragments were then sequenced and examined by western blotting. $BRCA1^{-/-}$ $53BP1^{-/-}$ double knockout cells were generated by BRCA1 sgRNA in $53BP1^{-/-}$ cells.

## Cell survival assay

Cell survival curves for HCT116 cells treated with HU were generated as described previously (*Katsube et al., 2011*). An appropriate number of cells was plated into 6-well plates with the indicated dose of HU. Cells were cultured for 9 to 14 days, and the colonies were stained with methylene blue and counted.

The cell survival assay for DT40 cells using MTT staining was performed as described previously (*Xu et al., 2010*). Briefly, 300–1000 cells were plated into each well of 96-well plates and incubated with a range of doses of HU or APH. After the cells were incubated 72 hr, the cells were pulsed with CellTiter 96 Aqueous One Solution Reagent (Promega) 4 hr. Cell viability was measured by a luminometer, and each dose point was measured in triplicate. For ICRF193, a density of 1500–3000 cells per well and a 48 hr incubation were used.

## Neutral comet assay

Neutral comet assay was carried out using a kit (Trevigen) as instructions. Cells were treated with HU (2 mM) for 12 hr, or CPT (1 μM) and Olaparib (1 μM) for 8 hr before harvest.

## Immunofluorescence and quantitative image-based cytometry (QIBC)

A modified immunofluorescence assay was performed as described previously (*Feng et al., 2016*). Briefly, HeLa or HCT116 cells were grown on poly-lysine-coated coverslips 24 hr before the experiments. The cells were washed with PBS once and then pre-extracted for 10 min at 4°C with CSK buffer (20 mM HEPES, pH 7.4, 100 mM NaCl, 300 mM sucrose, and 3 mM $MgCl_2$) containing 0.5% Triton-100. The cells were then washed three times with PBST (PBS with 0.1% Tween-20) and fixed with 3% PFA for 10 min at room temperature. After fixation, the cells were washed three times with PBST and blocked with 5% bovine serum albumin (BSA, Sigma) in PBS for 15 min. The cells were then incubated with the primary antibodies in PBS containing 1% BSA for 90 min. After washing, the cells were incubated with secondary antibodies diluted in PBS containing 1% BSA for 30 min. The cells were washed three times and mounted with ProLong Gold antifade reagent with DAPI (Invitrogen).

QIBC was performed as previously described (*Feng et al., 2016*; *Toledo et al., 2013*).

## DNA combing assay

A DNA combing assay was performed as described previously (*Davies et al., 2007*). Cells were labeled and treated as experimental designs, as indicated. Because 5 mM HU or 5 μM APH alone failed to completely block replication in HCT116 cells, a combination of 5 mM HU and 5 μM APH was used for blocking (*Figure 5—figure supplement 1A,B*). Then, the cells were trypsinized and diluted 1:4 with unlabeled cells at a concentration of $2.5 \times 10^5$ cells/ml. Then, 2.5 μl of cells was mixed with 7.5 μl of lysis buffer (200 mM Tris-HCl, pH 7.5, 50 mM EDTA and 0.5% SDS) on a clean glass slide. After 3–5 min, the DNA was allowed to slowly flow down along the slide by tilting the slides 15° horizontally. The slides were then air-dried, fixed in 3:1 methanol/acetic acid and refrigerated overnight. The slides were treated with 2.5 M HCl for 1 hr, neutralized in 0.1 M $Na_3B_4O_7$, pH 8.5, and rinsed three times in PBST (PBS buffer with 0.1% Tween-20). The slides were then blocked in blocking buffer (PBST buffer containing 1% BSA) for 20 min and incubated with rat anti-BrdU antibody (Abcam BU1/75, 1:200) in blocking buffer at 37°C for 1 hr. After three washes with PBST, the slides were incubated with Alexa Fluor 488-conjugated anti-rat (Molecular Probes, 1:200 dilution) for 45 min. After additional washes, the slides were incubated with mouse anti-BrdU (Becton Dickinson B44, 1:40) for 1 hr and then washed once with high-salt PBST (0.5 M NaCl) and three times with PBST. Then, the slides were incubated with Alexa Fluor 549-conjugated anti-mouse (Molecular Probes, 1:200 dilution) for 45 min. After three washes with PBST, the slides were mounted in SlowFade Gold antifade reagent (Invitrogen). The slides were imaged on a Zeiss Axiovert Microscope with a $100 \times$ objective.

## Statistics

Statistics was performed by two-tailed *t*-test or one-way ANOVA test. The data were normally distributed and the variance between groups being statistically compared was similar. No statistical methods or criteria were used to estimate sample size or to include/exclude samples. The investigators were not blinded to the group allocation during the experiments.

## Acknowledgements

We thank Weidong Wang, Daniel Durocher and Lee Zou for his advice and revisions to the manuscript. We thank the Core Facility of Life Sciences, Peking University for assistance with the imaging.

## Additional information

### Funding

| Funder | Grant reference number | Author |
| --- | --- | --- |
| National Basic Research Program of China | 2013CB911002 | Dongyi Xu |
| National Natural Science Foundation of China | 81672773 | Dongyi Xu |
| National Natural Science Foundation of China | 31661143040 | Dongyi Xu |
| National Natural Science Foundation of China | 31370836 | Rong Guo |

The funders had no role in study design, data collection and interpretation, or the decision to submit the work for publication.

### Author contributions

Yixi Xu, Shaokai Ning, Zheng Wei, Investigation, Writing—review and editing; Ran Xu, Xinlin Xu, Mengtan Xing, Investigation; Rong Guo, Conceptualization, Supervision, Funding acquisition, Investigation, Project administration, Writing—review and editing; Dongyi Xu, Conceptualization, Supervision, Funding acquisition, Writing—original draft, Project administration, Writing—review and editing

**Author ORCIDs**
Dongyi Xu (iD) http://orcid.org/0000-0001-5711-2618

**Decision letter and Author response**
Decision letter https://doi.org/10.7554/eLife.30523.019
Author response https://doi.org/10.7554/eLife.30523.020

## Additional files

**Supplementary files**
• Transparent reporting form
DOI: https://doi.org/10.7554/eLife.30523.016

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
