## [Decision Letter]

Thank you for submitting your article "53BP1 and BRCA1 control pathway choice for stalled replication restart" for consideration by *eLife*. Your article has been reviewed by three peer reviewers, and the evaluation has been overseen by a Reviewing Editor and Jessica Tyler as the Senior Editor. The reviewers have opted to remain anonymous.

The reviewers have discussed the reviews with one another and the Reviewing Editor has drafted this decision to help you prepare a revised submission.

Summary:

Xu et al. uncovered a new role for 53BP1 and BRCA1 DSB repair factors, genetically distinct from their classical DSB-repair role, in modulating fork progression, processing and restart upon transient or prolonged fork stalling. The data show that 53BP1 deficiency hampers the response to replication stress in vertebrates (both chicken and human cells), effects that are counteracted by BRCA1 deficiency. Thus, the authors show a role for BRCA1 in promoting active fork breakage and restart upon prolonged fork stalling, requiring PLK1 activation and SLX4-MUS81 mediated cleavage. The manuscript suggests a novel function of these factors in which the antagonistic relationship is conserved. Mechanistically, 53BP1 promotes a fast response to replication stress, and BRCA1 a slower response that requires fork cleavage by the SLX-MUS complex. In addition the authors propose a role for 53BP1 and RIF1 – but not Ku70, their functional partner in NHEJ – in promoting sustained fork progression and efficient restart early after fork stalling. The novelty of the study lies mostly in uncovering a direct role of BRCA1 in replication fork processing and replication restart, while the role of the alternative 53BP1/RIF1-dependent pathway is far less developed and understood. In general, the data provide novel insights into the pathways of choice for the restart of stalled replication forks. However, several concerns need to be addressed, as indicated below.

Essential revisions:

1) While the authors provide mechanistic investigations on the role of BRCA1 in fork restart upon prolonged stalling (SLX4-MUS81 binding, PLK1 involvement, role in MiDAS, epistasis with BIR factors), the mechanistic insight on the role of 53BP1/RIF1 in the early response to replication stress is limited to the analysis of a mutation which – among other possible effects – reduces 53BP1 interaction with BLM (it would be important to determine the genetic interaction between BLM and 53BP1 after HU treatment). However, there is no data assessing the relevance of this interaction in fork restart and also no direct assessment available from the literature confirming in vivo a role for BLM in replication fork remodeling (if anything, negative data from indirect experiments in Berti et al., NSMB 2013). Yet the authors propose in Results (subsection “The absence of BRCA1 rescues the replication stress-resistance of 53BP1-deficient cells”, third paragraph), Discussion (subsection “53BP1-RIF1 and BRCA1 have new functions in the restart of stalled replication forks”, first paragraph) and even in their graphical model (Figure 7), that this early role of 53BP1/RIF1 relates to promotion of fork remodeling. This interpretation is not supported by the data and could even be misleading in the field. In fact, the current data on BLM are inconclusive and should be removed from the manuscript. Considering the mechanistic insight provided on the other pathway studied here, it would be preferable to admit that the 53BP1/RIF1 early response to fork stalling is at this stage elusive and deserves further mechanistic investigation, rather than providing statements and models which could be easily proved wrong by further investigations in the field.

2) It is quite unclear throughout the manuscript what the authors mean for early stage vs. late stage of replication stress. Yet, this is a key distinction the authors make while designing and interpreting all experiments. While the PLK1 part of the work strongly suggests that direct fork processing and BIR-dependent restart require prolonged stalling and a sort of pre-mitotic stage, other data strongly argue that the BRCA1-dependent pathway must be operative also early upon fork arrest, namely 1) BRCA1 defects suppress 53BP1 defects upon short HU pulses (Figure 3), thus the BRCA1-dependent pathway competes with the 53BP1 mechanism early on; 2) relatively high – albeit not maximal – levels of PLK1 are detected already upon S phase entry (Figure 6). Although this important point may just need rephrasing throughout the manuscript, it is crucial that the authors build a coherent model around early and late events, consistent with the full set of their results.

3) Figure 2. The authors should conduct survival assays in HCT116 cells at multiple HU doses, as done for DT40 cells in Figure 1 and Figure 2. These experiments would better define the genetic interaction between BRCA1 and 53BP1 upon HU treatment in HCT116 cells, which are the cells primarily utilized in the manuscript.

4) While the effect of BRCA1 inactivation promoting the 53BP1-dependent pathway is quite clear, the opposite effect (53BP1 defect promoting slow fork restart, Figure 3) is far less convincing. In general, it would be beneficial for the authors to measure the efficiency of fork restart not only by the percentage of restarting forks, but also by the length of the restarted tracks (n.b.: this doesn't require new experimentation, but rather new careful assessment of their data set the authors should measure the length of the IdU tracts of the restarted forks in BRCA1- and 53BP1-deficient cells.). Activation of the BRCA1-dependent fork restart pathway should be visualized not only (or not necessarily) as delayed recovery of efficient fork restart, but also as shorter tracts at restarting forks, thus reflecting a delayed mechanism of fork restart (a key prediction of their model). Interestingly, this type of defect (delayed stalled fork restart) was recently reported for BRCA2-defective cells (Chaudhuri et al., Nature 2016) and very recent evidence has shown that delayed fork restart and successful S phase completion in these conditions does require fork processing by MUS81 (Lai et al., Nature Comms 2017). These data are of crucial importance for the model proposed in this paper and should be adequately introduced and discussed, especially if the new analysis suggested above were to yield important results in support of the proposed model.

5) Figure 4 shows a reduction of DSBs in BRCA1-/- cells that seems counterintuitive based on the previous literature showing an increase in chromosomal aberrations in BRCA1-deficient cells after HU treatment (Chaudhuri et al., Nature, 2016). The authors should determine whether the decrease in DSBs in BRCA1-/- cells after HU treatment results in reduced HU-induced chromosomal aberrations in HCT116 cells. It would also be important to perform these experiments upon shorter HU treatments, when not much DSBs are expected in wt cells (see also their own results in Figure 4). In these conditions the 53BP1 dependent restart mechanism should already be fully active and thus required to prevent fork breakage, maximizing a possible difference between 53BP1 wild type and mutant cells. These are also the ideal conditions to test whether increased breakage in 53BP1 defective cells are indeed dependent on BRCA1, thus testing directly the model proposed by the authors.

6) Figure 5. The authors measure the recruitment of the SLX-MUS complex to stalled replication forks by chromatin fractionation. However, the global changes observed in chromatin fractions may not be comparable to the changes observed at actual stalled forks. This is an essential point for the conclusions of the manuscript. To address it, the authors should perform iPOND or alternative approaches to monitor the direct recruitment of the SLX-MUS complex to nascent DNA in BRCA1-deficient cells upon replication stress. Indeed, these experiments should test whether increased loading of SLX4 is detected upon milder treatments and 53BP1 inactivation, where – according to the proposed model – the fork breakage pathway should be specifically boosted, another key prediction of their model and an ideal set-up to possibly consolidate it.

[Editors' note: further revisions were requested prior to acceptance, as described below.]

Thank you for resubmitting your work entitled "53BP1 and BRCA1 control pathway choice for stalled replication restart" for further consideration at *eLife*. Your revised article has been favorably evaluated by Jessica Tyler (Senior Editor), a Reviewing Editor, and two reviewers.

The manuscript has been improved but there are some remaining issues that need to be addressed before acceptance, as outlined below:

- The new Abstract does not look improved, in respect to the mentioned concerns. They start even earlier describing their own results, but the explanation of the open biological questions is still absent.

- The model still reports replication fork remodeling (fork reversal vs. fork restoration) as being directly controlled by 53BP1, with essentially the same emphasis as other transactions (i.e. fork breakage, BIR, SLX-MUS, PLK1), for which the authors provide solid lines of evidence. In the absence of any supporting data, graphical reference to fork remodeling is not justified and may generate confusion in the field. The model should be corrected, omitting any reference to fork remodeling, leaving this as a mere hypothesis for future work, simply mentioned in the Discussion.

- While the authors have referred (subsection “53BP1-RIF1 and BRCA1 have new functions in the restart of stalled replication fork”, last paragraph) to the two publications indicated by the referees in point 4, their citation in the text does not report the key findings in those papers that are relevant to this work, namely that BRCA2-defective cells display delayed restart of stalled forks (Ray Chaudhuri et al., 2016) and that this delayed restart is dependent on MUS81 (Lai et al., Nature Comms 2017; now also Lemacon et al., Nature Comms 2017, which got published in the meanwhile). Both points are crucially related to the conclusions of this study (including the authors' hypothesis on the role of BRCA2 in a cleavage-free restart pathway) and should thus be explicitly mentioned in the manuscript.

- The sentence "This phenomenon is quite similar as the case that the absence of 53BP1 only rescued HR in the BRCA1 deficient cells, but not the CtIP or XRCC2 deficient cells" is not grammatically correct.

- Delete "direct" in the following sentence "RIF1 is a downstream factor which may direct play an essential role".

- Figure 1, Figure 2, Figure 2—figure supplement 1, Figure 4—figure supplement 1. Change "Survive (%)" to "Survival (%)".

- Figure 5. The label of the last panel should be "53BP1", not "53BP".

---

## [Author Response]

Essential revisions:1) While the authors provide mechanistic investigations on the role of BRCA1 in fork restart upon prolonged stalling (SLX4-MUS81 binding, PLK1 involvement, role in MiDAS, epistasis with BIR factors), the mechanistic insight on the role of 53BP1/RIF1 in the early response to replication stress is limited to the analysis of a mutation which – among other possible effects – reduces 53BP1 interaction with BLM (it would be important to determine the genetic interaction between BLM and 53BP1 after HU treatment). However, there is no data assessing the relevance of this interaction in fork restart and also no direct assessment available from the literature confirming in vivo a role for BLM in replication fork remodeling (if anything, negative data from indirect experiments in Berti et al., NSMB 2013). Yet the authors propose in Results (subsection “The absence of BRCA1 rescues the replication stress-resistance of 53BP1-deficient cells”, third paragraph), Discussion (subsection “53BP1-RIF1 and BRCA1 have new functions in the restart of stalled replication forks”, first paragraph) and even in their graphical model (Figure 7), that this early role of 53BP1/RIF1 relates to promotion of fork remodeling. This interpretation is not supported by the data and could even be misleading in the field. In fact, the current data on BLM are inconclusive and should be removed from the manuscript. Considering the mechanistic insight provided on the other pathway studied here, it would be preferable to admit that the 53BP1/RIF1 early response to fork stalling is at this stage elusive and deserves further mechanistic investigation, rather than providing statements and models which could be easily proved wrong by further investigations in the field.

We appreciate this insightful suggestion. We agree that the current data on BLM are not strong enough. Thus, we removed these data from the manuscript as suggested. We will continue to study the mechanism of 53BP1/RIF1 on fork restart as an independent project in future.

2) It is quite unclear throughout the manuscript what the authors mean for early stage vs. late stage of replication stress. Yet, this is a key distinction the authors make while designing and interpreting all experiments. While the PLK1 part of the work strongly suggests that direct fork processing and BIR-dependent restart require prolonged stalling and a sort of pre-mitotic stage, other data strongly argue that the BRCA1-dependent pathway must be operative also early upon fork arrest, namely 1) BRCA1 defects suppress 53BP1 defects upon short HU pulses (Figure 3), thus the BRCA1-dependent pathway competes with the 53BP1 mechanism early on; 2) relatively high – albeit not maximal – levels of PLK1 are detected already upon S phase entry (Figure 6). Although this important point may just need rephrasing throughout the manuscript, it is crucial that the authors build a coherent model around early and late events, consistent with the full set of their results.

We apologize that we didn’t explain this model well. We fully agree that the BRCA1-dependent pathway is also operative early upon fork arrest. There is a balance between these two mutually exclusive pathways – the BRCA1-dependent pathway and the 53BP1-dependent pathway. In the early stage, the balance prefers to the 53BP1-dependent pathway, although the BRCA1-dependent pathway is also operative. In the late stage of replication stress, the increased PLK1 enhances the fork-cleavage activity of SLX-MUS complex, and the balance therefore tilts to the BRCA1-dependent pathway. Now we rephrased the manuscript and pointed out that the BRCA1-dependent pathway is enhanced/promoted, but not activated, when stress prolongs stalling or cells enter into pre-mitotic stage. We replaced the sentence “The 53BP1- and BRCA1-dependent pathways were activated in early and late stages of replication inhibition, respectively” as “The 53BP1- and BRCA1-dependent pathways mainly works in early and late stages of replication inhibition, respectively”, and the sentence “this BRCA1-dependent pathway is mainly activated in the late stage of replication stress” as “this BRCA1-dependent pathway mainly works in the late stage of replication stress”. We also added this point in the Discussion as “There is a balance between these two mutually exclusive fork restart pathways. In the early stage of replication stress, the balance favors to 53BP1-dependent pathway, although the BRCA1-dependent pathway is also operative. When replication stress is prolonged, the balance tilts toward the BRCA1-dependent pathway.”

3) Figure 2. The authors should conduct survival assays in HCT116 cells at multiple HU doses, as done for DT40 cells in Figure 1 and Figure 2. These experiments would better define the genetic interaction between BRCA1 and 53BP1 upon HU treatment in HCT116 cells, which are the cells primarily utilized in the manuscript.

We appreciate this reviewer’s suggestion. Now, we replaced it with new experiment as suggested (Figure 2).

4) While the effect of BRCA1 inactivation promoting the 53BP1-dependent pathway is quite clear, the opposite effect (53BP1 defect promoting slow fork restart, Figure 3) is far less convincing. In general, it would be beneficial for the authors to measure the efficiency of fork restart not only by the percentage of restarting forks, but also by the length of the restarted tracks (n.b.: this doesn't require new experimentation, but rather new careful assessment of their data set the authors should measure the length of the IdU tracts of the restarted forks in BRCA1- and 53BP1-deficient cells.). Activation of the BRCA1-dependent fork restart pathway should be visualized not only (or not necessarily) as delayed recovery of efficient fork restart, but also as shorter tracts at restarting forks, thus reflecting a delayed mechanism of fork restart (a key prediction of their model). Interestingly, this type of defect (delayed stalled fork restart) was recently reported for BRCA2-defective cells (Chaudhuri et al., Nature 2016) and very recent evidence has shown that delayed fork restart and successful S phase completion in these conditions does require fork processing by MUS81 (Lai et al., Nature Comms 2017). These data are of crucial importance for the model proposed in this paper and should be adequately introduced and discussed, especially if the new analysis suggested above were to yield important results in support of the proposed model.

We thank the reviewers for this insightful suggestion and appreciate the indicated literatures. We measured the length of the restarted tracks as suggested. Consistent with our model, although the percentage of restarting forks of the *53BP1^-/-^* cells is recovered to a level similar as that of wild-type cells when recover time is prolonged to 40 min, the length of the restarted tracks is significantly shorter (Figure 3), suggesting that the activated BRCA1-dependent pathway in the *53BP1^-/-^* cells is a delayed mechanism of fork restart. Now these data were included and discussed in the revised manuscript (subsection “53BP1 and BRCA1 promote the fast and slow kinetics restart pathways, respectively”, last paragraph). We also cited the two references and discussed their relevance in the context of our data (subsection “53BP1-RIF1 and BRCA1 have new functions in the restart of stalled replication forks”, last paragraph). In fact, we speculate that BRCA2 might play a role in the cleavage-free replication restart pathway together with 53BP1 (See discussion below about BRCA2).

5) Figure 4 shows a reduction of DSBs in BRCA1-/- cells that seems counterintuitive based on the previous literature showing an increase in chromosomal aberrations in BRCA1-deficient cells after HU treatment (Chaudhuri et al., Nature, 2016). The authors should determine whether the decrease in DSBs in BRCA1-/- cells after HU treatment results in reduced HU-induced chromosomal aberrations in HCT116 cells. It would also be important to perform these experiments upon shorter HU treatments, when not much DSBs are expected in wt cells (see also their own results in Figure 4). In these conditions the 53BP1 dependent restart mechanism should already be fully active and thus required to prevent fork breakage, maximizing a possible difference between 53BP1 wild type and mutant cells. These are also the ideal conditions to test whether increased breakage in 53BP1 defective cells are indeed dependent on BRCA1, thus testing directly the model proposed by the authors.

We apologize that we didn’t explain the relationship between HU-induced DSBs and chromosomal aberrations well. Our model predicts that HU-induced DSBs are intermediates of the cleavage-coupled BIR pathway (the BRCA1 and MUS81-dependent fork restart pathway), and they should not directly lead to chromosomal aberrations. In other words, the production of these DSBs should lead to resolving and restart of the stalled replication forks. Conversely, disruption of this pathway is expected to cause reduced HU-induced DSBs, increased levels of stalled replication forks and chromosomal aberrations. Consistent with this model, it has been reported that MUS81 deficient mouse ES cells or human HeLa cells showed a reduction of DSBs and an increase in chromosomal aberrations after HU treatment (Hanada K et al., Nat Struct Mol Biol. 2007, PMID: 17934473; Pepe A and West SC, Cell Rep. 2014, PMID: 24813886). Thus, there is no direct correlation between HU-induced DSBs and chromosomal aberrations. Our data have showed that MUS81 and BRCA1 are in the same restart pathway to promote the cleavage of the stalled forks (Figure 5). Thus, our results in Figure 4 that the *BRCA1^-/-^* cells showed a reduction of DSBs are not contradictory with the previous literature showing an increase in chromosomal aberrations in BRCA1-deficient cells after HU treatment (Chaudhuri et al., Nature, 2016).

We measured chromosomal aberrations in the *BRCA1^-/-^*HCT116 cells as suggested by the reviewers. Consistent with the previous literature (Chaudhuri et al., Nature, 2016), the *BRCA1^-/-^* HCT116 cells also show an increase of HU-induced chromosomal aberrations (See below). These phenomena in the *BRCA1^-/-^*HCT116 cells are similar to those of the MUS81 deficient cells described previously (Hanada K et al., Nat Struct Mol Biol. 2007, PMID: 17934473; Pepe A and West SC, Cell Rep. 2014, PMID: 24813886). They are also consistent with the notion that MUS81 and BRCA1 act in the same fork restart pathway. *53BP1^-/-^*cells did not show an obviously increase of chromosomal aberration after short HU treatment (see Author response image 1), although they did show an increase of DSBs (Figure 5). It is possible that HU-induced DSBs in the *53BP1^-/-^*cells were efficiently repaired by the BIR pathway before entering into M phase. Because these data are not important for the topic of our story, we didn’t include them in the revised manuscript. But if the editor and reviewers request us to include the data, we will be glad to add them.

**Author response image 1. respfig1:** BRCA1 suppresses chromosomal aberration upon replication stress. Sketch above the graphs delineates experimental design. The % of metaphase spreads with chromosomal aberrations in the indicated HCT116 cell lines was plotted. The mean and s.d. from two independent experiments are shown. ** P<0.01, * P<0.05.

6) Figure 5. The authors measure the recruitment of the SLX-MUS complex to stalled replication forks by chromatin fractionation. However, the global changes observed in chromatin fractions may not be comparable to the changes observed at actual stalled forks. This is an essential point for the conclusions of the manuscript. To address it, the authors should perform iPOND or alternative approaches to monitor the direct recruitment of the SLX-MUS complex to nascent DNA in BRCA1-deficient cells upon replication stress. Indeed, these experiments should test whether increased loading of SLX4 is detected upon milder treatments and 53BP1 inactivation, where – according to the proposed model – the fork breakage pathway should be specifically boosted, another key prediction of their model and an ideal set-up to possibly consolidate it.

We thank the reviewer for this insightful comment. We agree that the global changes observed in chromatin fractions may not reflect the changes at actual stalled forks. Our fractionation results are only consistent with our model, in which BRCA1 promotes the recruitment of the SLX-MUS complex to stalled forks, which are located on chromatin.

We also thank the reviewer for his/her suggestion to use iPOND or alternative approaches to monitor the direct recruitment of the SLX-MUS complex to nascent DNA. We have tried iPOND very hard, but were unable to overcome the technical difficulties to detect SLX4 and MUS81. We have sought the advice from Dr. David Cortez, who developed the iPOND technology. According to him, iPOND-immuno-blot is limited to detect small proteins that have very good antibodies available, such as RPA, Histone H3 and PCNA. For large proteins, the antigen epitopes may be damaged by formaldehyde crosslinking, so that they will be difficult to detect by immunoblotting. He said “I don’t think western blots after iPOND for BRCA1, 53BP1, SLX4, and Mus81 will be useful or practical. Doubtful that anyone could do them successfully. A better approach might by immunofluorescence imaging.” Thus, we followed his advice and used immunofluorescence imaging.

We used FANCD2 as marker of stalled forks and examined the recruitment of the SLX-MUS complex to stalled forks during MiDAS after mild replication stress (0.2 μM aphidicolin) treatment. This method has been broadly used (Minocherhomji S et al., Nature 2015, PMID: 26633632; Bhowmick R et al., Mol Cell. 2016, PMID: 27984745). Our results showed that BRCA1 promoted the recruitment of MUS81 to stalled forks, while 53BP1 suppressed it. These results are consistent with our model. Now we included these new results in the revised manuscript (Figure 5—figure supplement 1).

[Editors' note: further revisions were requested prior to acceptance, as described below.]

The manuscript has been improved but there are some remaining issues that need to be addressed before acceptance, as outlined below:- The new Abstract does not look improved, in respect to the mentioned concerns. They start even earlier describing their own results, but the explanation of the open biological questions is still absent.

Thank this reviewer for this suggestion. Now we have significantly changed the Abstract and included the explanations of the open biological questions that are the subject of this study. We specifically stated that “how many of these pathways exist in cells and how these pathways are selectively activated by different replication stress remain unclear.”

- The model still reports replication fork remodeling (fork reversal vs. fork restoration) as being directly controlled by 53BP1, with essentially the same emphasis as other transactions (i.e. fork breakage, BIR, SLX-MUS, PLK1), for which the authors provide solid lines of evidence. In the absence of any supporting data, graphical reference to fork remodeling is not justified and may generate confusion in the field. The model should be corrected, omitting any reference to fork remodeling, leaving this as a mere hypothesis for future work, simply mentioned in the Discussion.

Now we modified the model in Figure 7 and removed the graphical reference about fork remodeling.

- While the authors have referred (subsection “53BP1-RIF1 and BRCA1 have new functions in the restart of stalled replication fork”, last paragraph) to the two publications indicated by the referees in point 4, their citation in the text does not report the key findings in those papers that are relevant to this work, namely that BRCA2-defective cells display delayed restart of stalled forks (Ray Chaudhuri et al., 2016) and that this delayed restart is dependent on MUS81 (Lai et al., Nature Comms 2017; now also Lemacon et al., Nature Comms 2017, which got published in the meanwhile). Both points are crucially related to the conclusions of this study (including the authors' hypothesis on the role of BRCA2 in a cleavage-free restart pathway) and should thus be explicitly mentioned in the manuscript.

Now we explicitly mentioned these in the manuscript and cited the new literature as “Moreover, BRCA2-defective cells display delayed restart of stalled forks (Ray Chaudhuri et al., 2016) and this delayed restart is dependent on MUS81 (Lai et al., 2017; Lemacon et al., 2017), suggesting that BRCA2 may work in a MUS81-independent pathway to restart stalled forks. Thus, it will be interesting to study whether BRCA2 functions in the cleavage-free replication restart pathway together with 53BP1 in future.”

- The sentence "This phenomenon is quite similar as the case that the absence of 53BP1 only rescued HR in the BRCA1 deficient cells, but not the CtIP or XRCC2 deficient cells" is not grammatically correct.

We thank this reviewer for his/her careful reading. We re-wrote this sentence as “Similar phenomenon has been observed for repair of DSBs, in which the absence of 53BP1 rescued HR in only BRCA1-deficient cells, but not CtIP or XRCC2-mutant cells”.

- Delete "direct" in the following sentence "RIF1 is a downstream factor which may direct play an essential role".

We deleted “direct”.

- Figure 1, Figure 2, Figure 2—figure supplement 1, Figure 4—figure supplement 1. Change "Survive (%)" to "Survival (%)".

We changed them now.

- Figure 5. The label of the last panel should be "53BP1", not "53BP".

We corrected it now.